# Efficient thermal face recognition method using optimized curvelet features for biometric authentication

**Mona A. S. Ali**[1,2]*, **Mohamed Meselhy Eltoukhy**[3,4], **Fathimathul Rajeena P. P.**[1], **Tarek Gaber**[3,5]*

1 Computer Science Department, College of Computer Science and Information Technology, King Faisal University, Al Ahsa, Saudia Arabia, 2 Computer Science, Faculty of Computers and Artificial Intelligence, Benha University, Benha, Egypt, 3 Computer Science Department, Faculty of Computers and Informatics, Suez Canal University, Ismailia, Egypt, 4 Department of Information Technology, College of Computing and Information Technology at Khulais, University of Jeddah, Jeddah, Saudi Arabia, 5 School of Science, Engineering and Environment, University of Salford, Manchester, United Kingdom

* t.m.a.gaber@salford.ac.uk (TG); m.ali@kfu.edu.sa (MASA)

**Data Availability Statement:** All relevant data are within the manuscript and its Supporting information files.

## Abstract

Biometric technology is becoming increasingly prevalent in several vital applications that substitute traditional password and token authentication mechanisms. Recognition accuracy and computational cost are two important aspects that are to be considered while designing biometric authentication systems. Thermal imaging is proven to capture a unique thermal signature for a person and thus has been used in thermal face recognition. However, the literature did not thoroughly analyse the impact of feature selection on the accuracy and computational cost of face recognition which is an important aspect for limited resources applications like IoT ones. Also, the literature did not thoroughly evaluate the performance metrics of the proposed methods/solutions which are needed for the optimal configuration of the biometric authentication systems. This paper proposes a thermal face-based biometric authentication system. The proposed system comprises five phases: a) capturing the user's face with a thermal camera, b) segmenting the face region and excluding the background by optimized superpixel-based segmentation technique to extract the region of interest (ROI) of the face, c) feature extraction using wavelet and curvelet transform, d) feature selection by employing bio-inspired optimization algorithms: grey wolf optimizer (GWO), particle swarm optimization (PSO) and genetic algorithm (GA), e) the classification (user identification) performed using classifiers: random forest (RF), k-nearest neighbour (KNN), and naive bayes (NB). Upon the public dataset, Terravic Facial IR, the proposed system was evaluated using the metrics: accuracy, precision, recall, F-measure, and receiver operating characteristic (ROC) area. The results showed that the curvelet features optimized using the GWO and classified with random forest could help in authenticating users through thermal images with performance up to 99.5% which is better than the results of wavelet features by 10% while the former used 5% fewer features. In addition, the statistical analysis showed the significance of our proposed model. Compared to the related works, our system showed to be a better thermal face authentication model with a minimum set of features, making it computational-friendly.

**Funding:** The authors extend their appreciation to the Deputyship for Research and Innovation, the Ministry of Education in Saudi Arabia, for funding this research work (project number INST207.

**Competing interests:** No competing interests exist.

**Abbreviations:** 2D-DWT, Two-Dimensional Discrete Wavelet Transform; ANN, Artificial Neural Network; AUC, Area Under the Curve; CNN, Convolutional Neural Network; DB4, Daubechies 4; DCT, Discrete Cosine Transform; FDCT, Fast Discrete Curvelet Transform; GA, Genetic Algorithm; GWO, Grey Wolf Optimizer; GWO-TFFS, GWO-Thermal-Face-Feature Selection; KNN, K-Nearest Neighbour; LE, Local Energy; LSD, Local Standard Deviation; MCC, Matthews Correlation Coefficient; NB, Naive Bayes; PCA, Principle Component Analysis; PSO, Particle Swarm Optimization; RF, Random Forest; RLO, Random Linear Oracles; ROC, Receiver Operating Characteristic; SD, Standard-Deviation; SVM, Support Vector Machine; TFID, Terravic Facial IR Dataset; ZM, Zernike Moment.

# 1 Introduction

Facial recognition technology has become a topic of significant interest in recent years, with wide-ranging applications in fields such as security, surveillance, access control, and human-computer interaction. The rapid advancements in computer vision, machine learning, and deep learning have led to significant improvements in the accuracy and reliability of facial recognition systems, making them a promising solution for real-world scenarios. According to the International Biometrics + Identity Association (IBIA), facial recognition is one of the most widely used biometric modalities, with an estimated market size of 7.76 billion US dollars by 2025, with a compound annual growth rate (CAGR) of 15.3% from 2020 to 2025 [1]. The increased adoption of facial recognition systems can be attributed to their non-intrusive nature, ability to operate in real-time, and potential for high accuracy.

Biometric authentication, compared to traditional password or token-based authentication, can provide a higher level of security because it's much harder to fake or steal someone's biometric data than it is to guess or crack a password. Additionally, because biometric data is unique to each individual, it cannot be lost or forgotten in the same way that a password or token can be [2].

Biometrics refers to the automated recognition of an individual based on their unique physical or behavioral characteristics, such as fingerprints, facial features, iris patterns, or even voice. Biometric technology has gained popularity in recent years due to its ability to enhance security and improve usability by eliminating the need for passwords or PINs. In biometric authentication systems, recognition accuracy and security are critical factors that must be taken into account. The accuracy of biometric recognition depends on factors such as the quality of the sensors used, the algorithm used for matching biometric data, and the variability of biometric traits [3].

Biometric authentication systems use a variety of physiological and behavioral characteristics to identify individuals, such as fingerprints, iris patterns, facial recognition, voice recognition, and even gait analysis [4]. Each biometric characteristic has its own unique strengths and weaknesses, and the choice of which characteristic to use depends on the specific application and its requirements. For example, some characteristics may be more accurate or reliable than others, while others may be more convenient or user-friendly. It's important to note that no single biometric characteristic is expected to be effective for all applications. For example, fingerprints may work well for unlocking a smartphone, but may not be sufficient for secure access to a high-security area. The choice of which biometric characteristic to use must be carefully considered based on the specific needs of the application. Ultimately, the effectiveness of a biometric authentication system depends not only on the choice of biometric characteristics but also on the operational mode of the application and the properties of the biometric characteristic being used [4].

Deep learning algorithms have outperformed the traditional machine learning algorithms in many applications including cyber-physical health [5], detection of plant diseases [6], handwritten recognition [7], recognition of COVID-19 disease [8] and the price estimation [9]. Also, the deep learning-based models for Face recognition, e.g., [9–11], have made a noticeable advancement in face detection techniques, face matching algorithms. However, a recent study [12] found that performance (accuracy) enhancement comes at a huge computational expense required to train and manage deep-learning algorithms. This would hinder the adoption of deep learning-based face detection models.

Thermal imaging is proven to capture a unique thermal signature for a person, thus has been used in face recognition as in [10, 11, 13]. This thermal signature is primarily determined by the pattern of blood vessels that are present just under the surface of the skin, and these

patterns are unique to each individual. By using a thermal imaging sensor to capture this thermal signature, it is possible to create a digital representation of a person's face that can be used for identification and recognition purposes [14]. Thermal-based face recognition overcomes the challenges of visible face recognition in scenarios like surveillance carried out during the night when there is either very little or no light available to illuminate the faces, and occlusion problems [11]. It is also non-invasive and can be performed from a distance, making it more convenient for use in various applications. Face thermal imaging has several advantages over visual images including they are independent of lighting conditions and rely on radiations produced by the body rather than reflected light, making face recognition more robust against changing conditions [15, 16]. This is due to the fact that thermal imaging relies on thermal radiation rather than visible light [11].

The main challenge of thermal and visible-based face recognition systems is that they rely on their computational efficiency. These systems typically require significant computational resources to perform tasks such as face detection, feature extraction, and matching [14]. Such high computational costs hinder the adoption of face recognition technology in internet of things (IoT) applications such as security systems, access control systems, and surveillance systems. Selecting an optimal set of face features could help address this challenge.

From analysing the machine learning-based face recognition literature, e.g., [10, 17–19], a research gap was identified where most of the literature did not thoroughly analyse the impact of features selection (i.e., wavelet/curvelet features) on the accuracy and computational cost of face recognition. Also, the literature did not thoroughly evaluate the performance metrics of the proposed methods/solutions. The majority of the proposals only used accuracy as the evaluation metric. However, using the accuracy alone could be misleading as it treats false negatives and false positives as having the same costs, and it doesn't tell very much about the optimal configuration for the system.

Two main research questions were identified: 1- which is better (wavelet or curvelet transform) in extracting thermal face features to accurately and accurately recognise a user?, 2- which metaheuristic algorithms, GWO, GA and PSO, can select the minimum set of features that efficiently and accurately recognise users from their thermal face images?

This paper aims to propose an efficient and accurate thermal face recognition model for user authentication. This model first extracts the curvelet features from the thermal images, selects the optimal set of features using GWO with a novel proposed GWO-thermal-face-feature selection (GWO-TFFS) method, and then feeds the selected features to supervised lightweight machine learning classifier (the random forest classifier). In more detail, in the feature extraction phase, wavelet and curvelet features are used and their performance is compared in the selection phase, to give the most discriminative features with high accuracy without the need for high computational power using GWO, PSO and GA. In the classification phase, three learning strategies (instance-based learning (KNN), ensemble-based learning (RF) and probabilistic-based learning (NB)) are investigated to find out which will give the best performance for building thermal-face biometric authentication model.

The main contribution of this paper is summarized below.

1. Proposing a novel GWO-based feature selection method for thermal face recognition called GWO-thermal-face-feature Selection (GWO-TFFS) method. This is built after carefully investigating different features (i.e., wavelet and curvelet features) and using three different metaheuristic algorithms, GA, GWO and PSO, as feature selectors. The novelty of this method comes from the small percentage (6%-11%) of features selected by GWO to accurately identify the individuals using thermal images. This means that our proposed method could achieve 89%-94%- reduction rate, i.e., a highly efficient method.

2. Investigating three learning strategies,instance-based, ensemble-based and probabilistic-based learning, for the classification to build an efficient and accurate thermal-face biometric authentication model. This investigation, in conclusion, is expected to help practitioners and developers to decide which learning strategy will be the best for their applications.

3. Evaluating the proposed model using different types of performance metrics to ensure the effectiveness of this model. These metrics include accuracy, precision, recall, F-measure, and ROC area. Using these metrics, it was shown that our proposed model was comprehensively evaluated from different aspects to prove its effectiveness and quality.

4. Conducting statistical analysis of the proposed model to further show the significance between the different machine learning techniques employed in this study. This further helps practitioners make decisions (which techniques) based on solid evidence and avoid taking decisions based on random chance.

The rest of the paper is organized as follows. Section 2 summarizes and analyses the related work. Section 3 briefs the methods and approaches utilized in the proposed work. The proposed system is explained in Section 4. The experiment setup and evaluation and discussion are given in Section 5 and 6 respectively. Finally, the proposed work is concluded in Section 8.

## 2 Literature review

Face recognition using conventional visible spectrum heavily depends on illumination conditions, which can affect the accuracy of recognition systems. Therefore, many researchers have shifted towards thermal infrared recognition, which is less affected by illumination variations. A short survey of face recognition based on thermal images are summarized below.

Seal et al. [20] suggested a face recognition model using thermal images. They used the discrete wavelet transform (DWT) algorithm for feature extraction and dimensionality reduction. The experiments were conducted using a private database, and the results showed a recognition rate of 95%. On the other hand, when the model was tested using the Terravic Facial IR dataset(TFID), the recognition rate slightly decreased to 93%. This may be due to the differences between the private database and the TFID. Gaber et al. [13] proposed a thermal infrared recognition system that uses the segmentation-based fractal texture analysis (SFTA) algorithm to extract texture features from thermal images of human faces. The extracted features are then used to distinguish individuals by applying the random linear oracles (RLO) ensembles technique as a classifier. Their method has shown promising results in recognizing human faces using thermal images and has the potential for practical application in various fields, including security and surveillance systems.

In [21], the authors compared thermal-spectrum facial recognition methods. Local binary pattern (LBP) was used for feature extraction of face images. For dimensionality reduction, each feature set undergoes principle component analysis (PCA). Finally, a multi-layered feed-forward neural network and minimal distance classifier were used and achieved an accuracy of 95.09%. Their laboratory database and TFID were used for experiments. [22] introduced an image fusion technique that takes advantage of visible and thermal IR facial pictures. The wavelet transform was employed to extract features and RF classifier was applied to determine users from the thermal face pictures. RF classifier was applied to the fused face image and the results were 100% and 99.07% on IRIS benchmark face dataset and UGC-JU dataset, respectively. The model suggested by Ibrahim et al. [14] consists of four basic phases. Using the GWO method, the ideal superpixel values for the quick shift segmentation technique are

determined. Fractal texture analysis extracts the features, and rough set-based approaches choose the most discriminating features. AdaBoost is subsequently utilized to complete the categorization procedure. Their proposed method was evaluated using the TFID images. The classification accuracy obtained was 99%.

An approach has been proposed by Elbarawy et al. [23] to solve the expression recognition problem in thermal images using feature extractors and a classifier. The feature extractors used in their approach were the discrete cosine transform (DCT) and local energy (LE) filters, and the classifier used was the KNN. The experiments were conducted on the IRIS database, and the results showed that the approach using the LE filter achieved an accuracy of 90% in expression recognition. Interestingly, the same approach with additional features like PCA and local standard deviation (LSD) did not improve the accuracy of expression recognition. Ma et al. [24] suggested two techniques for identifying thermal facial images using local features. They started by incorporating feature types into multi-block LBP. In addition to a reasonably constant distribution of face temperature, they accommodate, around the reference, for a margin of error. This increased the resistance of the features to noise and their efficiency to face recognition in thermal pictures. The authors used AdaBoost to train cascade classifiers with various local feature types. Haar-like + HOG + AMB LTP achieved the highest f-score of 94% in all scenarios. Another thermal facial recognition method using gappy-principal component analysis and linear regression classifier was presented in [25]. Their technique achieved a 98.61% recognition rate on thermal face photos of the UGC-JU face database, which is set as the initial benchmark for performance identification for this database.

Mahesh et al. [26] proposed thermal IR face identification using Zernike moment(ZM) and MLPNN classifier. Their method was evaluated on the Terravic Facial IR Database in the front, right, and left poses and indoor and outdoor spaces. The rotation invariance and orthogonality of ZMs enhance capability representation. When ZM from orders 0 to 2 were concatenated to form a four-dimensional feature vector, the average recognition accuracy and the FAR rate was 89.5% and 0.365%, respectively. False alarms were lowered. The proposed model also classified images in regulated and real-world contexts with position variations. In [18] the authors present two infrared facial recognition systems using two databases. Backpropagation with one hidden layer classified all thermal and NIR database faces without extracting features. Traditional infrared facial recognition systems involve: Face data pre-processing, feature identification, dimensionality reduction to reduce data size, and a classifier to predict the test output. Backpropagation classified the test set by 100% accuracy with a few data. Rani et al. [17] introduced an infrared thermography(IT)-based technique that employs two-dimensional discrete wavelet transform (2D-DWT) to decompose thermal pictures of faces of various people and extract features. The dimensionality reduction of the extracted feature vector was done using PCA, and the selected features were assessed to find the most relevant feature vector. Artificial neural network(ANN) and support vector machine (SVM) were used to identify and classify people using a single feature vector. Using Terravic facial databases, ANN outperforms SVM where the latter achieved 100% accuracy.

The deep learning approaches were also suggested. The work in [19] presents a thermal facial recognition using convolutional neural network (CNN) architecture. The CNN automatically learns valuable features from raw data. Their CNN architecture outperforms HOG, LBP, and moments invariant on the RGB-D-T face database with 98% accuracy. Also, the study in [10] suggested a unique face thermal image feature-based approach that employed facial landmarks to construct a feature space based on standard-deviation (SD) and mean. Four models have been trained for the raw thermal, raw RGB, thermal feature and RGB feature pictures. The experiment employed 800 images for validation and training and 200 for testing. Random

**Table 1. Summary of related work using machine/deep learning techniques.**

| Year | Database | Features Extracted | Feature Selection | Classifiers | performance metrics |
|------|----------|--------------------|--------------------|-------------|---------------------|
| [10] | 10 subjects (1000) facial images | Statistical | N/A | GoogleNet | Accuracy from 85% to 99% based on body temperature |
| [14] | Terravic Facial IR | N/A | Rough Setbased techniqes | Adaboost | Accuracy-99%. |
| [17] | Terravic Facial IR | Statistical Features | PCA | SVM ANN Classifier | SVM 99.87% ANN 100% |
| [18] | Terravic Facial IR & NIR database | N/A | N/A | HOG-SVM Back-propagation Classifier | HOG-SVM 98.43% Back-propagation 100% |
| [19] | RGB-D-T based face recognition | N/A | N/A | CNN | Face recognition rate-98% |
| [21] | Terravic facial IR | Haar wavelet transform,LBP | N/A | ANN, Minimum distance | Accuracy -95.09% |
| [22] | UGC-JU IRIS face | invariant à-trous wavelet transform | N/A | Random Forest | Accuracy- 99.07% |
| [23] | IRIS | PCA LSD LE | N/A | KNN based on DCT | Accuracy PCA 60% LSD 80% LE 90% |
| [24] | UCH Thermal Face NVIE | Multi-Block LBP, HAAR-like, HOG, AMB,LTP | N/A | CASCAD Classifier | Fscore, Time, Precision and Recall |
| [25] | UGC-JU face | Gappy Principal Component Analysis | N/A | Linear Regression Classifier | Accuracy- 98.61% |
| [26] | Terravic Facial IR | Zernike moments | N/A | MLPNN Classifier | Accuracy 89.5% |

testing included 40 images. The results of the accuracy range from 85% to 99% based on body temperature.

The challenges of visible face recognition systems include position fluctuations, occlusion, ageing, illumination, and resolution [14]. Table 1, summarizing the literature survey above, three major limitations are identified. First, an efficient feature selection method for thermal face recognition is not suggested. Such a method should be selecting the smallest set of features such that it can produce accurate recognition results while not taking a high computational cost. Second, the literature did not thoroughly evaluate the performance of the proposed methods/solutions. The majority of the proposals only used accuracy as the evaluation metric. However, this could be misleading as it treats false negatives and false positives as having the same costs, and it doesn't tell very much about the optimal configuration for the system. Third, the deep learning technique gave excellent results in [10], but a recent study [12] found that performance (accuracy) enhancement comes at a huge computational expense required to train and manage deep-learning algorithms. Due to this, traditional businesses like European supermarkets have abandoned using deep-learning-based solutions [12]. This paper aims to propose a solution to address these problems.

## 3 Preliminary work

The following section presents a brief overview of the methods and algorithms utilized in the proposed study. The wavelet, curvelet, statistical features calculated, GWO, PSO and GA as feature selectors are described. In terms of the classification step, random forest, KNN and naive bayes classifiers are highlighted.

### 3.1 Wavelet transform

Wavelet transform is the most used method for calculating multi-resolution signal representations. This is because the wavelet transform approach may localize information in both the temporal and frequency domains [27]. Equation illustrates the wavelet transform calculation

formula (1).

$$\psi_{i,j} = \frac{1}{\sqrt{|i|}}\psi\left(\frac{x-j}{i}\right) \tag{1}$$

In Eq (1), $j$ and $i$ are the location parameter and the scaling factor, respectively. The primary goal of this transform is to approximate and interpret the signal using a collection of fundamental mathematical operations. An example of a function's wavelet transform is shown in Eq (2):

$$C_i(i,j) = \frac{1}{\sqrt{|i|}}\int_{-\infty}^{\infty} f(x)\psi\left(\frac{x-j}{i}\right)dx \tag{2}$$

The function $f(x)$ can be presented in the equation by its wavelet coefficients $C_f(i, j)$ where $i > 0, j \in R$. The discrete wavelet transform is performed by $i = 2^h, j = k2^h = ka$ for $k, h \in Z^2$. The method used to represent images is called a discrete curvelet transform. Image codes can edge more effectively using this method. This is due to the geometric component of the curvelet transform method. As a feature vector, the discrete curvelet transform approach's coefficients are employed. A fast discrete curvelet transform (FDCT) technique was the focus of research by [28].

The variables used in this study include $x$ as the spatial variable, $w$ as the frequency domain variable, and $r$ and $\theta$ as the polar coordinates in the frequency domain. Additionally, the study defines a pair of windows, namely $W(r)$ and $V(t)$. They serve, respectively, as radial and angular windows. While both $W$ and $V$ use real parameters supplied by $t \in [-1, 1]$ and $r \in (1/2, 2)$, respectively, all of the vectors are real, smooth, and positive, values. Non-negative real inputs are used by $W(r)$ and $V(t)$ respectively. Windows complies with the entrance requirements listed in [27]. Eqs (3) and (4) provide the following formulas for these windows:

$$\sum_{h=-\infty}^{\infty} W^2(2^h r) = 1, r \in \left(\frac{3}{4}, \frac{3}{2}\right) \tag{3}$$

$$\sum_{h=-\infty}^{\infty} V^2(t-1) = 1, t \in \left(-\frac{1}{2}, \frac{1}{2}\right) \tag{4}$$

For every $h \geq h_0$, a window of frequency in the Fourier domain, $U_h$ is defined by

$$U_h(r, \theta) = 2^{-\frac{3}{4}h} W(2^{-h}r) V\left(\frac{2^{\lfloor\frac{h}{2}\rfloor}\theta}{2\pi}\right) \tag{5}$$

Eq (5) uses the integer $\lfloor h/2 \rfloor$. The data from the radial window, $W$ and angular window, $V$ are also used to compute the polar wedge, $U_h$. The symmetrized form of Eq (5) can be derived by combining $U_h(r, \theta) + U_h(r, \theta + \pi)$ to produce real-valued curvelets. Similarly, the Fourier transform can be used to define the waveform $\varphi_h(x)$ as illustrated in Eq (6). $U_j(w_1, w_2)$ can be found using Eq (5), where the windows specified in the polar coordinate system are $w_1, w_2$. The major curvelet $\varphi_h$ is utilized in Eq (6) to represent all curvelets of scale $2^{-h}$ that can be obtained by rotating and translating $\varphi_h$. Now, we add rotation angles with $l = 0, 1, \ldots$ such that $0 \leq \theta \leq 2\pi$. These angles are $\theta_l = 2\pi.2^{\lfloor-h/2\rfloor}.l$. $k = (k1, k2) \in Z^2$, reveals the order of the translational parameters. The curvelets are thereby defined as a function of scale $2^{-h}$, orientation angle

*theta$_l$*, and position $x_k^{(h,l)} = R^{-1}\theta_l(k_1.2^{-h}, k_2.2^{-h/2})$

$$\varphi_{h,l,k}(x) = \varphi_h(R_{\theta_l}(x - x_k^{(h,l)}))$$ (6)

The rotation by $\theta$ radians can be calculated by $R_\theta$ and its inverse by $R_\theta^{-1}$

$$R_\theta = \begin{pmatrix} cos\theta & sin\theta \\ -sin\theta & cos\theta \end{pmatrix}, R_\theta^{-1} = R_\theta^T = R_{-\theta}$$ (7)

The inner product of an element $f \in L^2(R^2)$ and a curvelet $\varphi_{h,l,k}$, is defined as curvelet coefficient.

$$c(h, l, k) := \int_{R^2} f(x)\overline{\varphi_{h,l,k}(x)}dx$$ (8)

where $R$ denotes the real line. Length $\approx 2^{-h/2}$, width $= 2^{-h}$, i.e., width $\approx length^2$, known to be the anisotropy scaling relation or curve scaling law [28].

## 3.2 Wavelet and curvelet statistical characteristics

The proposed model utilizes ten descriptors or features to represent the coefficients in each sub-band of both wavelet and curvelet. These features are created based on the image's overall distribution of grey levels. The ten features in our proposed strategy are statistical ones and are formally detailed below [29]. **Mean** is the mathematical average of a set of numeric data, $z_1, z_2, ..., z_m$. [30]:

$$\bar{z} = \frac{1}{m}(z_1 + z_2 + ... + z_m) = \frac{1}{m}\sum_{i=1}^{m}z_i$$ (9)

**Variance** is a measure that illustrates the spread of the histogram, indicating the extent to which the grey levels ($I[x, y]$) differ from their mean ($\bar{I}[x, y]$). Also, the variance provides insight into the range of a random variable's values. The square of the deviations from the mean are averaged to create the variance. The square root of variance is known as the **standard deviation** [30].

$$Var = \frac{1}{m-1}\sum(I[x,y] - \bar{I}[x,y])^2$$ (10)

$$STD = \sqrt{\frac{1}{m-1}\sum(I[x,y] - \bar{I}[x,y])^2}$$ (11)

The third moment $\mu_3$ of a standardized random variable is referred to as **skewness**, which determines the degree of asymmetry in the histogram around the mean. The clarity of the histogram is measured using **kurtosis**, denoted as $\mu_4$. If significant skewness and kurtosis are present, the data may not be considered normal [29]. Eqs (12) and (13) define skewness and

kurtosis as:

$$S = E\left[\left(\frac{X - \mu}{\sigma}\right)^3\right] \tag{12}$$

$$K = E\left[\left(\frac{X - \mu}{\sigma}\right)^4\right] \tag{13}$$

**Energy** is a statistical feature that represents the sum of the squared magnitude of coefficients in a sub-band. It is often used to measure the amount of signal content in a particular sub-band. The following formula [30] can be used to compute Energy:

$$Energy = \sum_{x}^{R}\sum_{y}^{C} I^2[x, y] \tag{14}$$

The measure of **entropy** is employed to assess the degree of randomness in an image, with smoother images resulting in lower entropy and rougher images resulting in higher entropy. The formula used to calculate entropy is presented below [30]:

$$Entropy = -\sum_{x}^{R}\sum_{y}^{C} I[x, y] \log I[x, y] \tag{15}$$

The highest value in the provided matrix is referred to as the **maximum** value. The calculations are as follows. Where $R$ and $C$ represent rows and columns respectively:

$$R, C$$

$$Max = max\ I[x, y] \tag{16}$$

$$x, y$$

The consistency of the element distribution in the different shades of grey is measured using the term "homogeneity." Its value ranges from 0 to 1. A smoother texture image results from a value that is near to 1, and so on. The definition of an image's **homogeneity** in mathematics is [29]:

$$Homogeneity = \sum_{x}^{R}\sum_{y}^{C} \frac{1}{1 + (x - y)^2} \cdot I(x, y) \tag{17}$$

**Moment** is a property of a picture that is utilized widely in image classification and pattern recognition [29].

$$moment = \sum_{x}^{R}\sum_{y}^{C} \frac{I(x, y)}{(x - y)^2} \tag{18}$$

### 3.3 Grey wolf optimizer (GWO)

Mirjalili et al. [31] proposed the grey wolf optimizer (GWO), which is a type of metaheuristic optimization approach. The algorithm's leadership pyramid and hunting strategy are similar to those of grey wolves, therefore the name. A grey wolf pack typically has five to twelve members and is organized into the social classes alpha, beta, delta, and omega. Here, alpha is regarded as the alpha dog. All of the pack's crucial decisions are made by the alpha. The alpha

wolf is supported by the beta wolf in decision-making and other activities as it moves down the food chain. Delta wolves also referred to as sub-ordinate wolves, are dominant over omega wolves yet work closely with alpha and beta wolves.

The alpha and beta wolves provide guidance and safety to the pack, while the omega wolves are regarded as disposable members or scapegoats. Nonetheless, omega wolves play a critical role in scouting, warning the pack of potential threats, and defending against external threats. The symbols used to represent alpha, beta, delta, and omega are $\alpha$, $\beta$, $\delta$, and $\omega$, respectively. To use the GWO for addressing the feature selection problem, the study in [32], proposed a competitive binary grey wolf optimizer. In our study, the GWO was modeled as given in Algorithm 1 which illustrates how GWO was used as a feature selection technique.

**Algorithm 1** The pseudo-code for GWO for feature selection:

```
1: Initialize the population:
  • Generate random binary strings of length N (where N is the number
of features) to form the initial population.
2: Define the fitness function: KNN (classification accuracy)
  • Evaluate the fitness of each individual in the population using a
classification accuracy of KNN.
3: Initialize the positions of the alpha, beta, and delta wolves.
  • Select three individuals with the highest fitness scores and
assign them as alpha, beta, and delta wolves.
4: Update the positions of the wolves.
  • Update the position of each individual in the population. using
the following equations:
```

- For alpha wolf: x_alpha = x_alpha + A * D_alpha

- For beta wolf: x_beta = x_beta + A * D_beta

- For delta wolf: x_delta = x_delta + A * D_delta

- For the rest of the population: x_i = (x_alpha + x_beta + x_delta + x_i) / 4 + rand() * (x_alpha−x_beta)

```
  • where x_i is the position of the i th individual, A is the step
size, and D_alpha, D_beta, and D_delta are the distance vectors of the
alpha, beta, and delta wolves, respectively.
5: Evaluate the fitness of the new population.
  • Calculate the fitness of each individual in the population using
the fitness function.
6: Repeat steps 4-5 for a certain number of iterations or until a sat-
isfactory solution is found.
7: Select the best feature subset.
  • Select the individual with the highest fitness score as the best
feature subset.
8: Return the best feature subset.
```

## 3.4 Particle swarm optimizer (PSO)

Kennedy and Eberhart [33] introduced PSO as a swarm-based algorithm that imitates animal social behavior such as bird flocking and fish schooling. Each potential solution is represented as a fast-moving particle, similar to a swarm of birds, traversing the problem space. The particle combines a portion of its best historical position and current location, determined by one or more agents of the swarm, with random disturbances to determine its future path through the search space. Once all particles have moved, the next iteration begins. PSO's main advantage is that it requires fewer tuning parameters. However, the high-dimensional search space

can slow down the convergence to the global optimum [34]. In our paper, the PSO was modeled as a feature selector as given in Algorithm (2).

**Algorithm 2** The pseudo-code for a particle swarm optimization (PSO) algorithm for feature selection:

```
1: Initialization: Generate a population of particles, each represent-
ing a set of features.
2: Define the fitness function: KNN (classification accuracy)
  • Evaluate the fitness of each particle in the population using a
classification accuracy of KNN.
3: Initialization: Set the best position (set of features) and fitness
value found so far for each particle and the entire swarm.
4: Repeat steps 5-8 for a number of iterations or until maximum number
of iterations.
5: Update particle velocity: For each particle, update its velocity
based on its previous velocity, its distance to its best position and
the swarm's best position, and two tuning parameters (inertia weight
and acceleration coefficients).
6: Update particle position: For each particle, update its position by
adding its new velocity to its current position.
7: Evaluation: Evaluate the fitness of each particle's new position.
8: Update best position: For each particle, update its best position
and fitness value found so far, and update the swarm's best position
and fitness value if the particle's fitness is better than the swarm's
best fitness.
9: Return the best position found, which represents the best subset of
features for the classification task.
```

## 3.5 Genetic algorithms optimizer (GAO)

Genetic algorithms simulate biological evolution and model evolutionary processes to solve problems. Genetic algorithms can evolve sophisticated and intriguing structures in a simple computational framework. Population genetics is the basis for genetic algorithms. The population is randomly constructed, with each individual represented by a bit string that represents a potential answer to the problem at hand. Some individuals are fitter than others due to genetic variation within the population. Selection biases the next group of potential solutions based on these differences. By copying and eliminating successful individuals, selection creates a new population with some differences among its members. During this copying process, mutation, crossover, and other bit-string changes may occur. Mutation and crossover procedures create new samples with a better-than-average likelihood of being good by changing the existing group of good individuals. After many generations of appraisal, selection, and genetic operations, the population's fitness improves, and its members become better "solutions" to the fitness function's problem [35]. In our paper, the GAO was modeled as a feature selector as given in Algorithm (3).

**Algorithm 3** The pseudo code for Genetic Algorithm (GA) for feature selection:

```
1: Initialize a population of chromosomes with random values within
the search space. each representing a set of features.
2: Define the fitness function: KNN (classification accuracy)
3: Evaluation: Evaluate the fitness of each chromosome by measuring
the classification accuracy rate of KNN on the subset of features it
represents.
4: Selection: Select a subset of chromosomes to mate based on their
fitness.
5: Crossover: Create new chromosomes by combining the features of the
selected chromosomes.
```

```
6: Mutation: Randomly mutate some of the chromosomes by changing the
value of some of their genes.
7: Evaluation: Evaluate the fitness of each particle's new position.
8: Repeat steps 3-6 for a number of generations (maximum number of
generations).
9: Return the best chromosome found, which represents the best subset
of features for the classification task.
```

## 3.6 Classification: Random forests (RF)

Random forest method is a strategy for ensemble learning that combines many classifiers to improve a model's performance. It is a decision tree-based supervised machine learning technique and is employed for both regression and classification issues [36]. The random forest's ensemble notion states that merging a number of ineffective classifiers could result in a high classification rate. Multiple decision trees' outputs were blended by random forest to increase prediction accuracy. The classification stage in this study was carried out using 10-fold cross-validation to evaluate the efficacy of the suggested methodology.

## 3.7 Classification: K-nearest neighbors (KNN)

The K-Nearest Neighbors classifier is a straightforward, yet powerful, classification algorithm that falls under the category of instance-based learning or lazy learning algorithms. During the training phase, KNN only stores the data without constructing any explicit models. To classify a new instance, KNN calculates its similarity with its K-nearest neighbors in the training set. KNN offers several benefits, such as being simple to implement, tolerant of noisy data, and capable of handling multi-class classification. However, KNN also has some drawbacks, such as its sensitivity to the choice of K and the distance metric employed.

## 3.8 Classification: Naive bayes (NB)

The Naive Bayes classifier is a well-known machine learning and data analysis algorithm that uses the Bayes theorem of conditional probability to make predictions. It is a probabilistic model that assumes the features are independent of each other, hence the name "naive". Despite this simplifying assumption, the algorithm exhibits impressive performance in a broad range of applications, such as text classification, spam filtering, and sentiment analysis. One of the Naive Bayes's significant advantages is its simplicity and speed, making it an efficient algorithm for handling large datasets. Additionally, it requires minimal training data and can work well even with high-dimensional data. While NB may not be the optimal algorithm for every problem, it is still a popular choice for many machine learning tasks.

## 4 Proposed approach

The proposed thermal face-based biometric authentication system, as shown in Fig 1, would have five phases. The first phase involves capturing the user's face image using a thermal camera. In the second phase, an optimized superpixel-based segmentation technique [14] was used to extract the ROI of the face from the captured image. In this technique, GWO was employed to locate the optimal superpixel settings of the quick-shift segmentation algorithm. The GWO algorithm was used to search for the optimal values of the quick-shift parameters that result in the best segmentation performance of the face thermal images. The results in [14] have shown that the optimized superpixel-based segmentation technique has produced the best ROI which has improved the classification rate [14]. Based on these results, superpixel-based segmentation technique was used to extract the

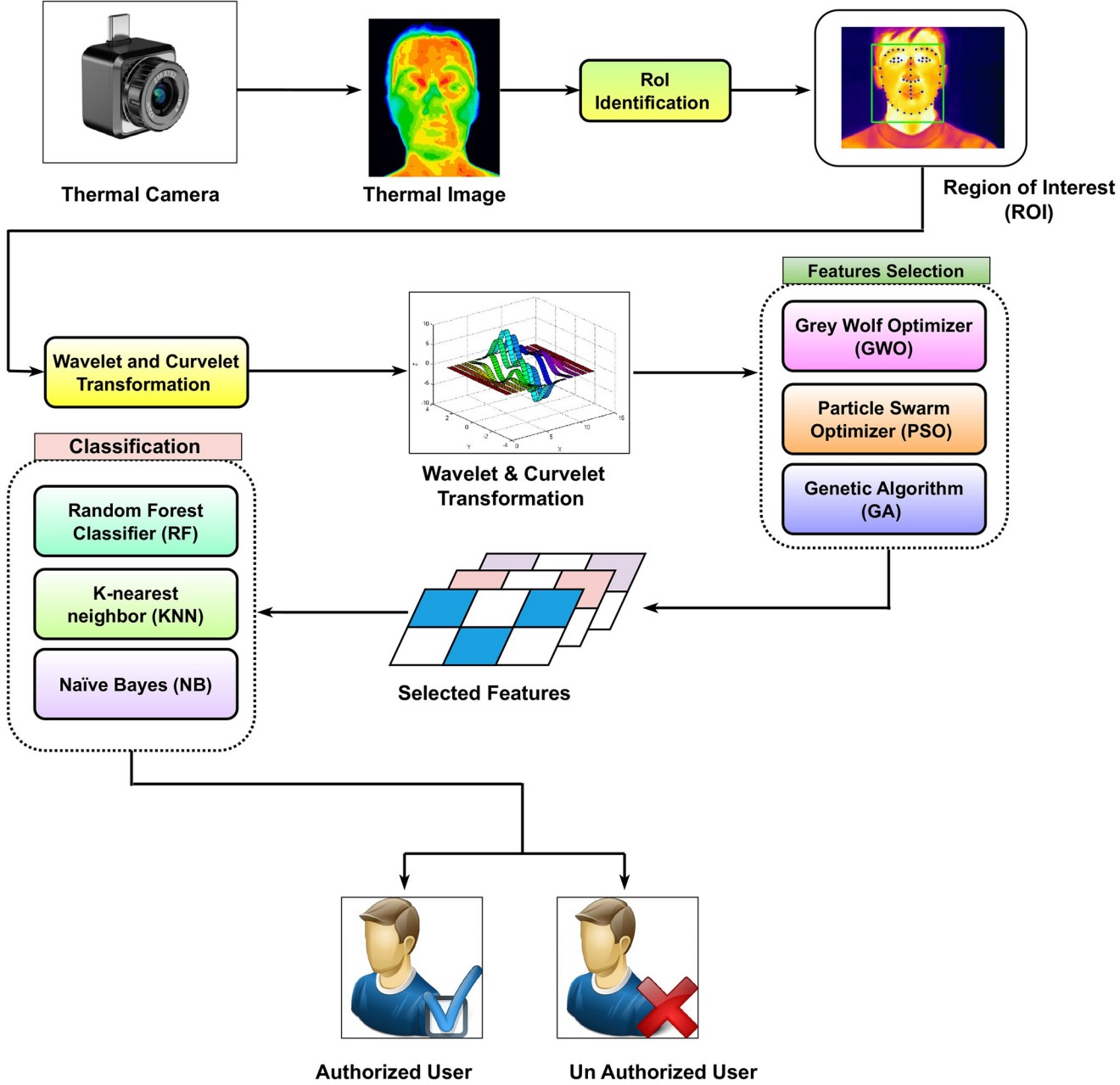

**Fig 1. Schematic architecture of the proposed approach.**

ROI from the thermal image to minimize the processing time and to extract the most related facial features.

In the feature extraction phase, the curvelet and wavelet are used to transform the ROI into the frequency domain. The aim is to compare these two techniques in extracting a set of accurate features that enable a high accuracy of users' authentication.

The curvelet and wavelet transforms are mathematical techniques used to analyze signals and images in the frequency domain. Both transforms decompose an image into several sub-bands, which contain different frequency components of the image. In the case of the curvelet

transform, the decomposition is achieved by dividing the image into small wedges, each of which is represented by a set of curvelet coefficients. These coefficients capture the local frequency information of the image in a directional manner. Ten statistical features, as described in Section 3.2, are then computed from each curvelet sub-band or wedge, which are used to construct feature vectors for the region of interest. The curvelet transform is designed and utilized using four scales and sixteen angles, which means that the input data is being decomposed into a set of sub-bands with different levels of frequency and orientation. Each sub-band is then further decomposed into a set of small regions called wedges, which capture the local curvature and texture of the data. In this case, there are 81 wedges in total, which are formed by combining the four scales and sixteen angles. From these wedges a feature vector is constructed by concatenating all of the wedge coefficients into a single vector of length 810.

The wavelet transform, on the other hand, decomposes the image into sub-bands using a set of basis functions known as wavelets. These basis functions are designed to capture the local frequency information of the image in a multiscale manner. Statistical features are also calculated from each wavelet sub-band and used to construct feature vectors for each ROI of the thermal face images, as in Section 3.2.

The efficiency of the Daubechies 4 (DB4) wavelet has been established according to the findings in [37, 38]. As a result, we utilised this wavelet in our proposed technique to decompose the ROI image into four distinct levels. Hence, the (DB4) wavelet was used to decompose the image i.e, the ROI into four levels, each level consisting of four sub-bands: approximation (A), horizontal (H), vertical (V), and diagonal (D), resulting in a total of 16 sub-bands for each image. From each sub-band, ten features are calculated. Thus, a total of 160 features being extracted from each ROI. These features are used to produce the wavelet feature vector, which is used for further analysis or classification purposes.

The feature selection phase is proposed to boost learning accuracy and quality of classification performance by only selecting the most discriminative features while removing irrelevant, redundant and noisy ones. Metaheuristic-based feature selection techniques have shown to be efficient in this space [39, 40]. So, in this paper, two types of metaheuristic algorithms (evolutionary or swarm) were used as feature selection techniques. For the evolutionary type, we used GA while for the swarm type, we used PSO and GWO. The main goal was to identify the best set of features which improve thermal face recognition accuracy while minimizing the computational time.

In the classification phase, the best set of features, selected by the metaheuristic technique, was given to a supervised classifier to identify users given their thermal images. Based on three different learning strategies (instance-based learning (K-Nearest Neighbour), an ensemble-based learning (Random forest) and probabilistic based learning (Naive Bayes) were used. Each of these three classifiers was applied to three sets of features produced by GA, GWO, and PSO. The best results were then reported to build the final model. See the results section for more details.

## 5 Experimental setup

This section describes the dataset and performance metrics used in evaluating the proposed method. It also presents parameters of the used classifiers as well as experiments environment settings.

### 5.1 Dataset

The public dataset, Terravic Facial infrared (IR) [41], was adopted in the evaluation of the proposed models. The dataset contains 4000 images for 20 different classes. In each of the classes, there is a total of 200 black-and-white images. The photos are JPEG files that are 8 bits

grayscale and 320 pixels by 240 pixels in size. Each class is meant to represent a unique individual. Every individual possesses certain images that can be viewed in a variety of configurations (front, left, right; indoors/outdoors; glasses) to represent different life scenarios. As a result of the corruption that occurred in the fifth and sixth classes, this evaluation only utilised a total of 18 classes (i.e., 3600 images were used).

## 5.2 Performance metrics

The effectiveness of the proposed method was assessed utilizing a series of commonly-used benchmark metrics that were derived from the confusion matrix obtained during the classification stage. These metrics include:

Accuracy is a metric that quantifies the percentage of accurate predictions made during the classification process.

$$Accuracy = \frac{TP + TN}{TP + TN + FP + FN} \tag{19}$$

where TP is the number of true positive instances, TN is the number of true negative instances, FP is the number of false positive instances, and FN is the number of false negative instances.

Precision is a metric that determines the ratio of true positive outcomes to all the predicted positive outcomes.

$$Precision = \frac{TP}{TP + FP} \tag{20}$$

Recall, also known as Sensitivity or True Positive Rate, is a performance metric that quantifies the ability of a model to correctly identify the positive instances.

$$Recall = \frac{TP}{TP + FN} \tag{21}$$

The F-Measure is a performance metric which combines the values of precision and recall into a single score. It is computed as the harmonic mean of precision and recall and can be expressed mathematically as follows:

$$F_{Measure} = \frac{2*(Precision*Recall)}{(Precision + Recall)} \tag{22}$$

The Receiver Operating Characteristic (ROC) curve's Area Under the Curve (AUC) is a widely-used metric that assesses the effectiveness of classification models. The ROC curve illustrates the true positive rate (sensitivity) versus the false positive rate (1-specificity) at various classification thresholds. An AUC score of 1.0 indicates that the model has a perfect performance, whereas an AUC score of 0.5 suggests a random guess. The ROC curve provides a comprehensive summary of the classifier's overall performance.

The Matthews Correlation Coefficient (MCC) is a performance metric utilized to assess the binary classifier's effectiveness. It depends on the entries in the confusion matrix and measures the correlation between the observed and predicted binary classifications. The MCC is mathematically expressed as:

$$MCC = \frac{TP*TN - FP*FN}{\sqrt{(TP + FP)*(TP + FN)*(TN + FP)*(TN + FN)}} \tag{23}$$

The MCC can take values from -1 to 1, where -1 indicates a completely incorrect classifier, 1 indicates a perfect classifier, and 0 indicates a random classifier. The MCC is a robust metric

that provides a balanced evaluation of the performance of a classifier, taking into account both false positive and false negative errors. These metrics provide a comprehensive evaluation of the performance of the proposed method.

### 5.3 Experiments environment setting

All the experiments in this paper were executed under the following settings. The specification of the used PC was: Intel(R) Core(TM) i7–10700 CPU @ 2.90GHz with 8.00 GB RAM and the experiments were implemented using MATLAB R2019a under Windows 10. Also, all the experiments in the scenarios below were conducted in 10 independent runs because the optimization process starts with random solution(s) in Meta-Heuristic. Hence, it is basically affected by the parameter's initialization. Therefore, ten independent runs were conducted. The average of the ten runs was then calculated and reported.

A good training approach for a model with a dataset must be found. The model should have enough instances to train on without over-fitting it, but if there aren't enough, the model won't be properly trained and will perform poorly when tested [42, 43]. Using the k-fold cross-validation, with a large value of k (i.e., k approaching the number of instances in the dataset) or leave-one-out cross-validation can help ensure that the model is evaluated on as much of the data as possible. This can provide a more accurate estimate of the model's performance than a simple train-test split, as each data point is used for both training and validation [42]. It was justified in [42] that the k-fold cross-validation is better to be used over hold-out validation. Therefore, in our case, k-fold cross-validation has been used to evaluate the performance of the proposed models.

Unlike, splitting the dataset into 70% for training and 30% for testing a model, in a k-fold cross-validation the dataset is split into k subsets, or folds, of approximately equal size. The model is then trained on k-1 folds, and the remaining fold is used for validation/testing. This process is repeated k times, with each fold used as the validation set exactly once. The results of the k-folds are then averaged to obtain an overall estimate of the model's performance. This approach is useful because it allows for a more accurate estimation of the model's performance, as each data point is used both for training and validation at some point in the process [42, 43].

### 5.4 Parameter settings

This section presents an explanation of the values of the parameters used in each classifier and feature selection method. An experiment was conducted to determine the best value of KNN parameter, k. The results, given in Table 2, show that the best value of k is 5 which was then used in all experiments of KNN below.

Random Forest parameters are given in Table 3. The Max_Depth parameter can affect the performance of random forest, so we run an experiment aimed at finding the best value and giving the best results. It was observed that the best value of Max_Depth is to make it as long as possible, meaning that Nodes are expanded until all leaves are pure, as shown in Table 4. Consequently, this value was adopted for all subsequent experiments involving random forest. The parameters values of the feature selection algorithms used in the proposed method are given in Tables 5–7.

## 6 Results and analysis

To evaluate the proposed model, we designed four experimental scenarios.

**Table 2. Impact of K value on the KNN performance.**

| Number of K Nearest Neighbours | Accuracy | Precision | Recall | F-Measure | MCC | ROC Area | PRC Area |
|---|---|---|---|---|---|---|---|
| K = 1 | 0.991 | 0.991 | 0.991 | 0.991 | 0.991 | 0.996 | 0.984 |
| K = 2 | 0.988 | 0.989 | 0.988 | 0.988 | 0.988 | 0.999 | 0.995 |
| K = 3 | 0.991 | 0.991 | 0.991 | 0.991 | 0.99 | 0.999 | 0.998 |
| K = 4 | 0.991 | 0.991 | 0.991 | 0.991 | 0.99 | 0.999 | 0.998 |
| **K = 5** | **0.993** | **0.993** | **0.993** | **0.993** | **0.992** | **1** | **0.999** |
| K = 6 | 0.992 | 0.992 | 0.992 | 0.992 | 0.991 | 1 | 0.999 |
| K = 7 | 0.992 | 0.993 | 0.992 | 0.992 | 0.992 | 1 | 0.999 |
| K = 8 | 0.99 | 0.991 | 0.99 | 0.99 | 0.99 | 1 | 0.999 |
| K = 9 | 0.99 | 0.991 | 0.99 | 0.99 | 0.99 | 1 | 0.999 |
| K = 10 | 0.99 | 0.99 | 0.99 | 0.99 | 0.989 | 1 | 0.999 |

1. Minimum set of feature identification: The goal of this investigation is to determine which category of meta-heuristic algorithms (GA, PSO, or GWO) is the most effective for performing feature selection.

2. Best learning strategy of classifiers: The purpose of this scenario is to study the most accurate learning strategy among instance-based, ensemble-based and probabilistic-based learning approach, for thermal face image classification.

3. Best classifier performance: The objective of this scenario is to analyze and compare the performance of different metaheuristic algorithms such as GA, GWO, and PSO, in terms of selecting the optimal set of features (from wavelet and curvelet) that can achieve the highest classification accuracy using the best classifier identified in Scenario 2.

4. Most efficient model: This scenario aims to compare between the computational time required for the model when using GA, GWO, or PSO as feature selector.

In the following subsections, each scenario is described in detail.

## 6.1 Scenario 1: Identification of the minimum set of features

The aim of this scenario was to investigate which type of meta-heuristic algorithms (evolutionary or swarm) would be the best feature selection technique. For the evolutionary type, we used GA while for the swarm type, we used PSO and GWO. The main goal was to identify the best set of features which improve face recognition accuracy while minimizing the computational time. To achieve this goal, GA, PSO, and GWO were used as feature selection techniques on the features extracted by the wavelet and curvelet transformation.

**Table 3. Parameter settings of random forest.**

| Parameter | Explanation | Value applied |
|---|---|---|
| N Estimators | Number of trees in the forest | 100 |
| Max depth | the longest path between the root node and the leaf node | Unlimited: Nodes are expanded until all leaves are pure |
| Min samples split | Minimum number of samples requiredto split an internal node | 2 |
| Min samples leaf | Minimum number of samples required to be at a leaf node | 1 |
| Max leaf nodes | Maximum no of leaf nodes generated | Unlimited number of leaf nodes |

**Table 4. Impact of changing the Max_Depth value on the performance.**

| Max_Depth | TP Rate | Precision | Recall | F-Measure | MCC | ROC Area |
|---|---|---|---|---|---|---|
| 2 | 0.888 | 0.901 | 0.888 | 0.885 | 0.883 | 0.992 |
| 4 | 0.96 | 0.96 | 0.96 | 0.96 | 0.957 | 0.998 |
| 6 | 0.98 | 0.98 | 0.98 | 0.98 | 0.979 | 0.999 |
| 8 | 0.988 | 0.988 | 0.988 | 0.988 | 0.988 | 1 |
| 10 | 0.992 | 0.992 | 0.992 | 0.992 | 0.992 | 1 |
| **Unlimited** | **0.993** | **0.993** | **0.993** | **0.993** | **0.993** | **1** |

**Table 5. Parameter settings of grey wolf optimizer.**

| Parameter | Explanation | Value applied |
|---|---|---|
| Population size | The number of wolves in the search space at each iteration | 10 |
| Search space | The range of possible solutions for the problem being optimized | The number of features in the given space |
| Iteration | The number of iterations or generations that the algorithm will run | 100 |

**Table 6. Parameter settings of particle swarm optimizer.**

| Parameter | Explanation | Value applied |
|---|---|---|
| inertia weight (w) | balance the global exploration and local exploitation | 0.9 |
| best personal solutions (C1) | It determines the weight given to the particle's personal best position in determining its next move | 2 |
| best global solution (C2) | It determines the weight given to the best position found by the entire swarm in determining its next move | 2 |

**Table 7. Parameter settings of genetic algorithms.**

| Parameter | Explanation | Value applied |
|---|---|---|
| Population Size | The number of individuals in each generation of the GA | 10 |
| Crossover Rate (CR) | The probability of performing a crossover operation between two parents to create offspring. | 0.8 |
| Mutation Rate (MR) | the probability of introducing random changes to an individual's genetic material during reproduction. | 0.01 |

The experiment was conducted in ten independent steps because the optimization process in metaheuristic starts with random solutions. Hence, it is basically affected by the parameter's initialization. Therefore, ten independent experiments were conducted, and the average of the ten runs was calculated. The average accuracy of each optimizer was recorded and compared as well as the time CPU taken was also calculated and compared. In addition, the percentage of the selected features for each method was reported. This was to show the amount of feature reduction each algorithm (GA, PSO, and GWO) can make. The summary of the results of this scenario is given in Table 8.

From these results, presented in this table, the following remarks can be made. Firstly, the results of all algorithms (GA, PSO, and GWO) using curvelet features were better than using

**Table 8. Feature selection by GA, GWO and PSO.**

| | GA as a feature selection technique | | | | | | | |
|---|---|---|---|---|---|---|---|---|
| Feature Selection | Curvelet | | | | Wavelet | | | |
| | Accuracy | Time(s) | Features | Percentage | Accuracy | Time(s) | Features | Percentage |
| GA_1 | 98.87% | 165.47295 | 262 | 32.35% | 86.17% | 30.758998 | 20 | 12.50% |
| GA_2 | 98.58% | 177.517759 | 239 | 29.51% | 86.18% | 21.09969 | 13 | 8.13% |
| GA_3 | 98.15% | 164.092158 | 259 | 31.98% | 86.45% | 23.162714 | 19 | 11.88% |
| GA_4 | 98.86% | 160.75248 | 245 | 30.25% | 86.18% | 25.076062 | 20 | 12.50% |
| GA_5 | 98.26% | 176.462746 | 288 | 35.56% | 86.12% | 25.816473 | 19 | 11.88% |
| GA_6 | 98.86% | 168.719418 | 266 | 32.84% | 86.22% | 24.018041 | 21 | 13.13% |
| GA_7 | 98.72% | 173.59331 | 278 | 34.32% | 86.87% | 22.558828 | 15 | 9.38% |
| GA_8 | 98.72% | 164.100608 | 260 | 32.10% | 86.43% | 23.460321 | 18 | 11.25% |
| GA_9 | 98.80% | 164.149605 | 266 | 32.84% | 86.32% | 25.025994 | 23 | 14.38% |
| GA_10 | 98.29% | 163.861135 | 274 | 33.83% | 86.70% | 21.401498 | 13 | 8.13% |
| **Average** | **98.61%** | **167.8722169** | **263.7** | **32.56%** | **86.36%** | **24.2378619** | **18.1** | **11.31%** |
| | Grey Wolf as a feature selection technique | | | | | | | |
| Feature Selection | Curvelet | | | | Wavelet | | | |
| | Accuracy | Time(s) | Features | Percentage | Accuracy | Time(s) | Features | Percentage |
| GWO_1 | 99.12% | 78.096678 | 198 | 24.44% | 86.12% | 11.520921 | 11 | 6.88% |
| GWO_2 | 98.15% | 68.817696 | 151 | 18.64% | 87.15% | 11.485777 | 10 | 6.25% |
| GWO_3 | 99.28% | 64.037209 | 155 | 19.14% | 86.20% | 11.75062 | 16 | 10.00% |
| GWO_4 | 99.18% | 58.503095 | 129 | 15.93% | 86.26% | 10.380232 | 9 | 5.63% |
| GWO_5 | 99.25% | 55.753692 | 114 | 14.07% | 87.11% | 11.380216 | 14 | 8.75% |
| GWO_6 | 99.18% | 57.869282 | 121 | 14.94% | 86.28% | 11.327911 | 13 | 8.13% |
| GWO_7 | 98.24% | 51.652055 | 91 | 11.23% | 86.43% | 11.352478 | 16 | 10.00% |
| GWO_8 | 99.16% | 63.736463 | 133 | 16.42% | 86.39% | 12.245027 | 15 | 9.38% |
| GWO_9 | 98.21% | 66.524285 | 158 | 19.51% | 86.26% | 12.940165 | 25 | 15.63% |
| GWO_10 | 98.52% | 55.55739 | 94 | 11.60% | 87.05% | 11.542802 | 10 | 6.25% |
| **Average** | **98.83%** | **62.0547845** | **134.4** | **16.59%** | **86.53%** | **11.5926149** | **13.9** | **8.69%** |
| | Particle Swarm as a feature selection technique | | | | | | | |
| Feature Selection | Curvelet | | | | Wavelet | | | |
| | Accuracy | Time(s) | Features | Percentage | Accuracy | Time(s) | Features | Percentage |
| PSO_1 | 98.72% | 116.530399 | 343 | 42.35% | 87.45% | 21.404575 | 57 | 35.63% |
| PSO_2 | 98.72% | 118.344307 | 336 | 41.48% | 83.50% | 21.054404 | 65 | 40.63% |
| PSO_3 | 98.72% | 111.705514 | 346 | 42.72% | 87.22% | 18.085649 | 45 | 28.13% |
| PSO_4 | 98.86% | 120.335204 | 370 | 45.68% | 86.88% | 20.592737 | 59 | 36.88% |
| PSO_5 | 98.72% | 108.794494 | 331 | 40.86% | 86.32% | 22.839912 | 67 | 41.88% |
| PSO_6 | 98.58% | 119.825416 | 351 | 43.33% | 86.31% | 21.259194 | 62 | 38.75% |
| PSO_7 | 98.58% | 117.606615 | 374 | 46.17% | 87.31% | 20.604929 | 62 | 38.75% |
| PSO_8 | 98.15% | 122.397801 | 367 | 45.31% | 87.42% | 23.949061 | 70 | 43.75% |
| PSO_9 | 98.58% | 118.55687 | 370 | 45.68% | 86.04% | 20.910869 | 59 | 36.88% |
| PSO_10 | 98.95% | 121.054354 | 381 | 47.04% | 86.18% | 20.708786 | 64 | 40.00% |
| **Average** | **98.66%** | **117.5150974** | **356.9** | **44.06%** | **86.46%** | **21.1410116** | **61** | **38.13%** |

wavelet features for thermal face images. The curevelet-based results were better than the wavelet-based results with about %12 on average (98% for curvelet and 86% for wavelet). This superiority of curvelet transformation is attributed to its ability to divide the image into 16 angles, as opposed to wavelet transformation which only divides the image into a few horizontal, vertical, and diagonal directions. Thus, the curvelet can extract more discriminating features than wavelet can.

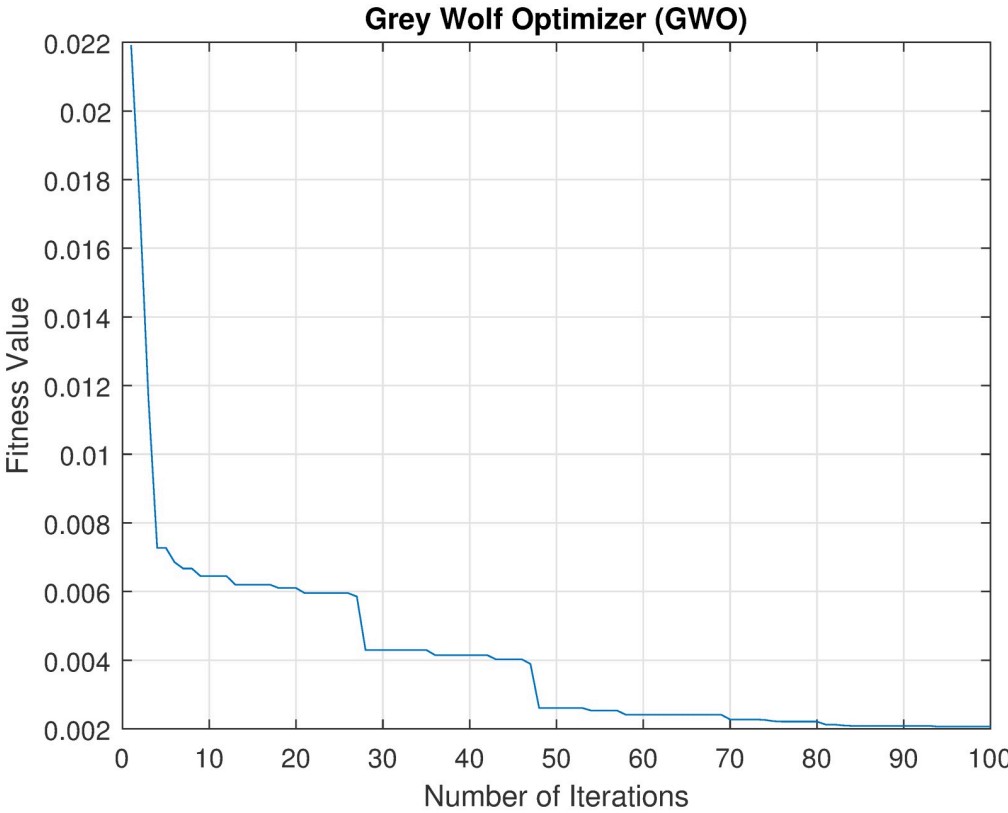

**Fig 2. The convergence of GWO to find the best set of features.**

Secondly, the average reduction rate of the wavelet features was much better higher than that of the cuverlet features. For the wavelet features, the average reduction rate was 89.69%, 91.31%, and 61.87% for GA, GWO, and PSO, respectively. For the curevelt features, the average reduction rate was 67.54%, 83.41%, and 55.94% for GA, GWO, and PSO, respectively. It was also noticed the GWO has achieved the highest reduction rate for both type of features while giving the highest accuracy too.

Thirdly, the results in Table 8 indicate that GWO was the best in terms of accuracy, time consumption, and percentage of the selected features (i.e., selected the minimum set of features giving the highest accuracy). The average accuracy achieved by the GWO was 98.83%, while the average time taken was 62.05 seconds. Furthermore, the average number of features selected was 134.4, which constituted only 16.59% of the total features. On the other hand, the GA gave an average accuracy of 98.61%, an average time consumption of 167.87 and an average number of 263.70 for selected features. These results demonstrate that the GWO method outperforms the GA in terms of both accuracy and time consumption, i.e., the swarm methods is better than the evolutionary method.

**6.1.1 Convergence analysis.** Convergence of metaheuristics algorithms is a very important indicator of their performance. Thus, an experiment was conducted to investigate the relation between fitness value against the number of iterations of each of the used algorithms (GA, GWO, and PSO). Note that in this study, the fitness function was to minimize this error rate which is calculated for the classification accuracy of the model on the dataset, i.e., maximizing the accuracy. The best way to illustrate this relation is by plotting it. Figs 2–4 show the convergence of each method in finding the best solution. The x-axis represents the number of

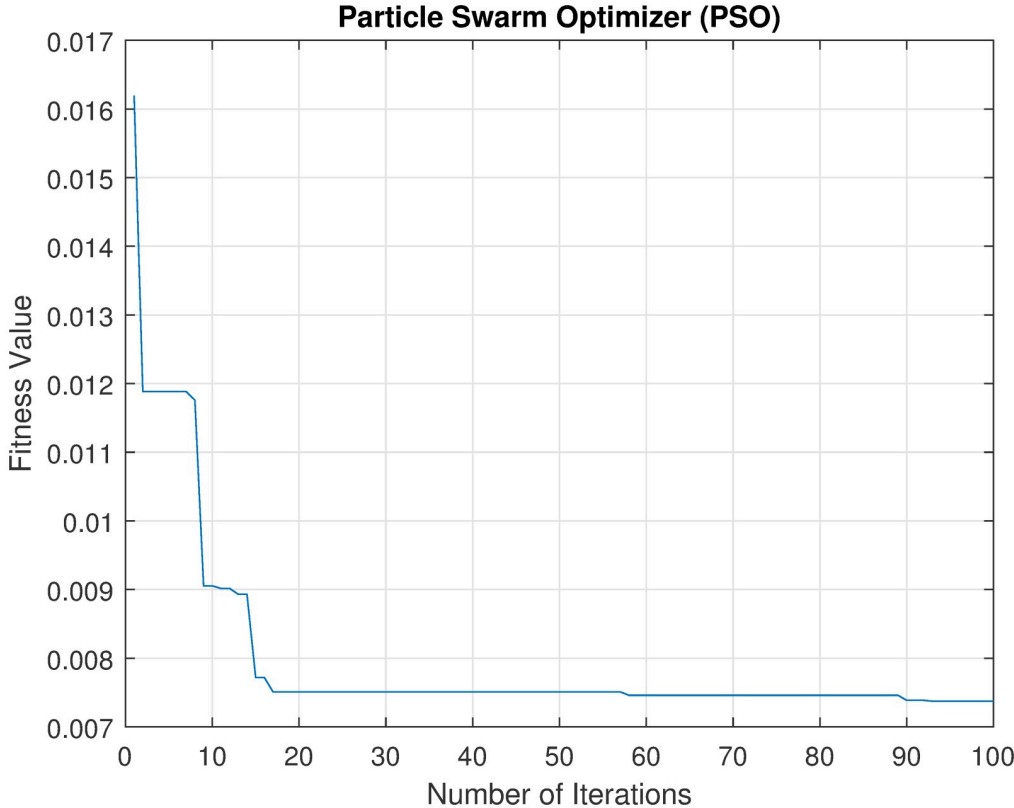

**Fig 3. The convergence of PSO to find the best set of features.**

iterations, and the y-axis represents the best fitness value achieved by the algorithm at that iteration.

From these figures, it can be seen that as the number of iterations increases, there is a decreasing trend in the fitness value, indicating that the algorithm is making progress towards finding a good solution. When comparing the three convergence curves, it can been noticed that the GWO is the best solution with a fitness value of 0.002, while PSO and GA found solutions with fitness values of 0.0074 and 0.0043, respectively. These results suggest that GWO outperforms both PSO and GA as feature selection methods.

Overall, the results of this scenario support the conclusion that the GWO-based method is a superior feature selection method in terms of accuracy, time consumption, and the number of selected features. Also, the best accuracy, 99.28% was achieved by the set of features, GWO_3. Furthermore, the results showed that our proposed method could achieve 89% (using GWO_7 and GWO_10) and 94% (using GWO_4 and and GWO_10) reduction rate by GWO-based method for curvelet features and wavelet features respectively. Such high reduction rate of the features will not require high computation cost to recognise users. This means that our proposed GWO-based features selection method for thermal face recognition could supports limited resources, e.g., IoT applications.

### 6.2 Scenario 2: Best learning strategy of classifiers

Given the best-selected features (i.e., curvelet features from Scenario 1), in Scenario 2 it was aimed to investigate the best-supervised learning strategy. Those strategies are instance-

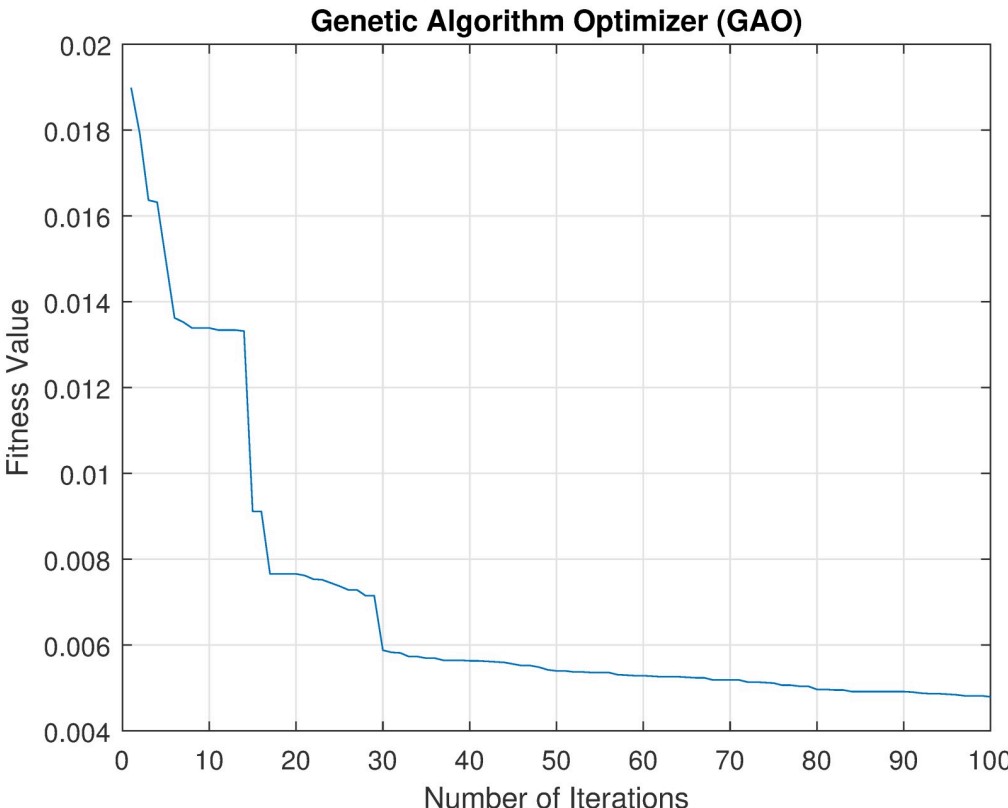

**Fig 4. The convergence of GA to find the best set of features.**

based learning, ensemble-based learning and probabilistic-based learning, and the classifiers are KNN, RF and NB, respectively. From the first experiment, it was proved that the GWO was the best feature selector for both wavelet and curvelet features in terms of the minimum number of features and the highest accuracy rate. So, in Scenario 2, we designed two sub-experiments (wavelet-based features and curvelet-based features) in which the three classifiers were applied to the sets of features produced by GWO. In both sub-experiments, the classifiers, KNN, RF, and NB were applied to wavelet and curvelet features selected by GWO and the results were reported as given in Figs 5 and 6. The presented figures illustrated the average of ten runs for each classifier with best selected features from GWO method.

From Figs 5 and 6, two main remarks can be noticed. Firstly, the RF classifier outperformed the other classifiers in all performance metrics. RF achieved average results of 99.4% with some set of features achieving above 99.6% by all evaluation metrics. In other words, the ensemble-based learning strategy was better than the instance-based learning and probabilistic-based learning in identifying users through their thermal faces. Secondly, like in Scenario 1, the curvelet features showed to give classification results better than the wavelet features in all evaluation metrics although the wavelet features can identify the users using a number of features less than that are needed by curvelet features. In conclusion, the results of all classifiers can be sorted as follows, RF, KNN and NB. The results of this scenario motivated us to further apply the random forest classifier over the features selected by three optimization algorithms as presented in Scenario 3.

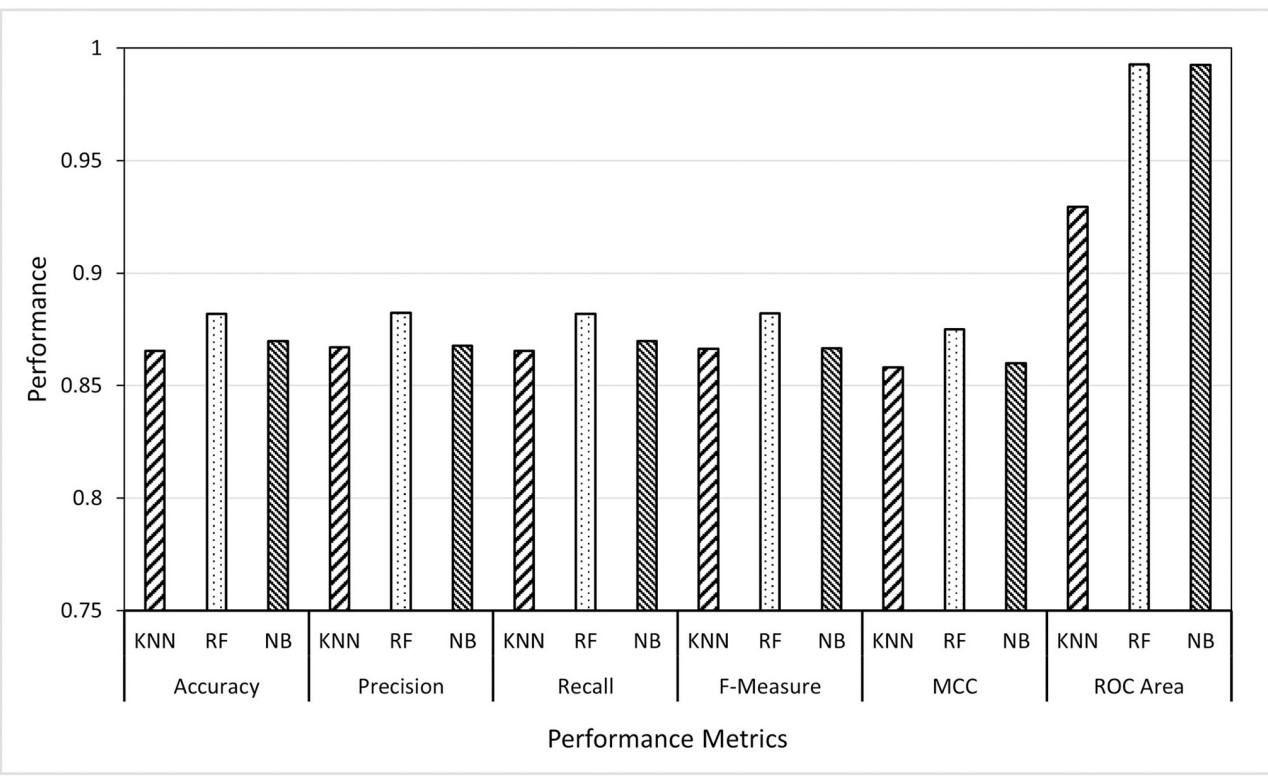

**Fig 5. Results of KNN, RF and NB classifiers using wavelet features.**

## 6.3 Scenario 3: Best classifier performance

The aim here is to investigate which metaheuristic algorithm among GA, GWO and PSO would select the set of features (from wavelet and curvelet) which can give the highest classification performance using the best classifier identified from Scenario 2 (i.e., RF). In other words, this scenario does not consider the computational time nor the minimum number of features as in Scenario 1, the only concerned with the accuracy and other performance metrics. To achieve this, in Scenario 3, we designed two sub-experiments (wavelet-based features and curvelet-based features) in which, the features selected by three metaheuristic algorithms (GA, GWO and PSO) were given to the RF classifier (best classifier from Scenario 2) and the results of these two experiments were reported as given in Figs 7 and 8.

The obtained results, from the curvelet features, showed that the average of the performance metrics of PSO features is outperforming the other two feature selectors, i.e., GWO and GA. The GWO and GA came second and third, respectively. Although the GWO was not the best in accuracy and other measures, it gave comparable results with the smallest average of the number of features needed to accurately identify the users. On the other hand, the area under curve in ROC showed an equal performance for the three FS methods reaching 100%. This means that the three-feature selection (FS) methods being compared in the study performed equally well in terms of their ability to discriminate between the different classes. An AUC of 100% indicates perfect discrimination between all users, which means that the classifier is able to correctly authenticate all users without any errors.

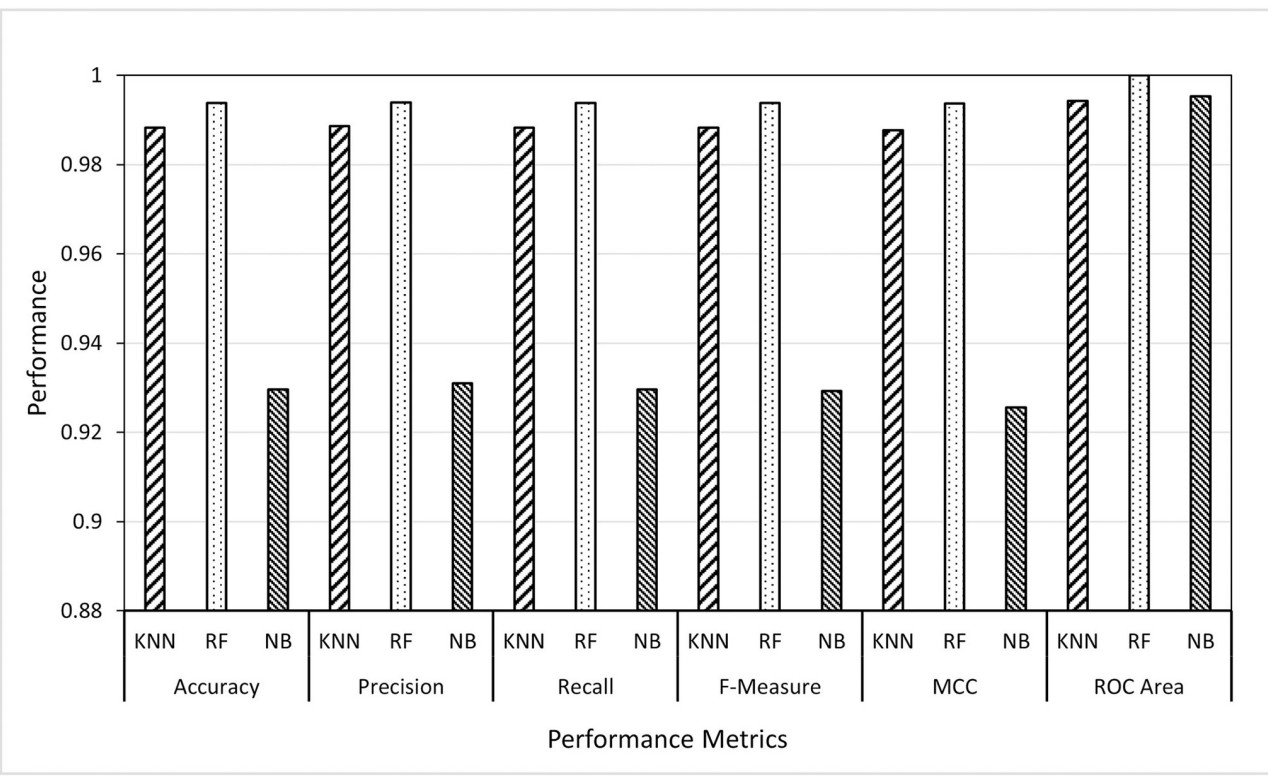

**Fig 6. Results of KNN, RF and NB classifiers using curvelet features.**

## 6.4 Scenario 4: Most efficient model

This scenario conducted to measure the relation between the number of iteration of each algorithm (GA, GWO, and PSO) and the CPU time needed to reach the highest accuracy (i.e., to find out the most efficient model). As the random forest classifier with the curvelet features gave the highest results, see Table 8, we used them in measure the computational time of using GA, GWO, and PSO as feature selection. Fig 9 illustrates that the GWO outperformed both GA and PSO in the computational time consumed. This confirms the results presented and discussed in Scenario 1. This scenario provides additional insights into the computational efficiency of the algorithms, which is an important factor to consider when selecting an optimization algorithm for a particular application. The findings suggest that GWO may be a good choice for applications where computational efficiency is a critical factor.

## 6.5 Statistical analysis

In general, the significance analysis provides a rigorous evaluation of the proposed methods and helps to ensure the reliability of the results obtained. By using appropriate statistical tests, the authors would be able to draw robust conclusions about the performance of the different methods and their relative strengths and weaknesses. In this study, due to the diversity of the obtained results (see results of Scenario 1–3), the authors conducted a significance analysis to evaluate the differences between the proposed methods. This analysis was conducted in two phases. In the first phase, the authors investigated whether the data follow a normal distribution or not. This was done by using the Shapiro-Wilk test. The results of this test showed that

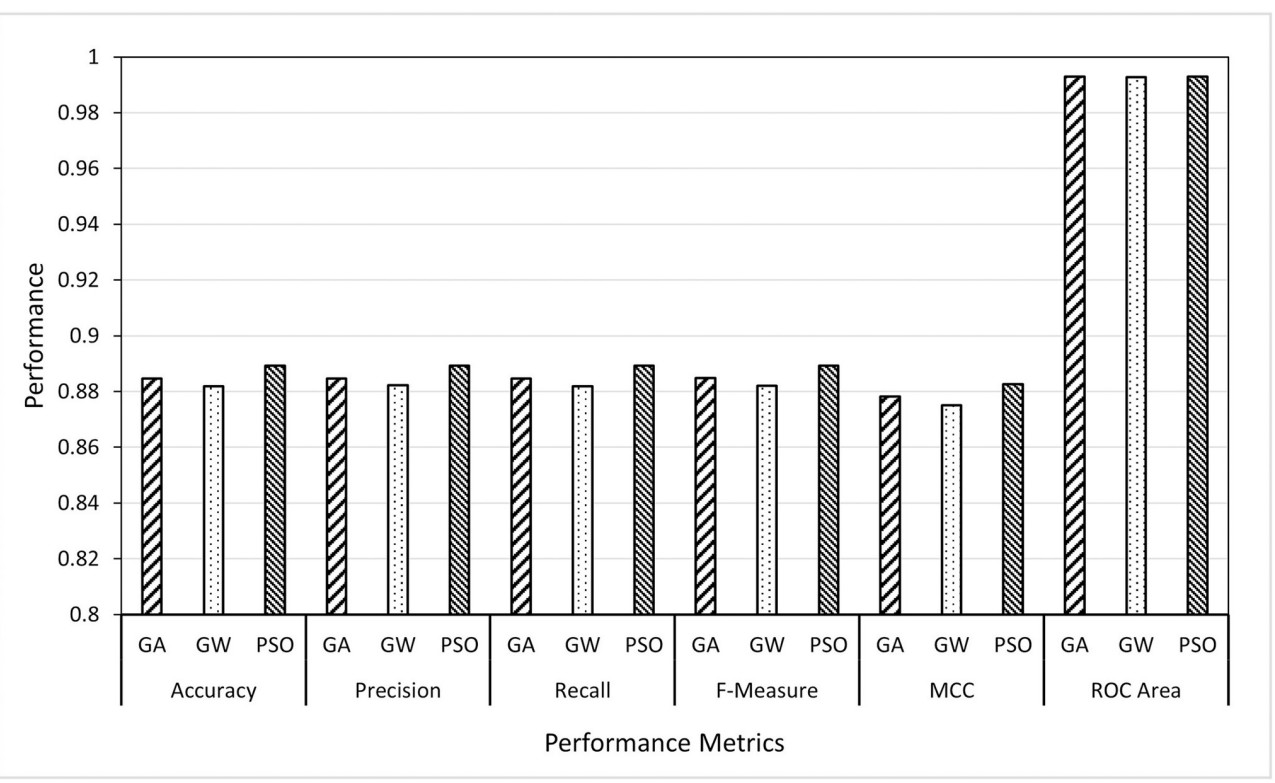

**Fig 7. Comparing GA, GWO and PSO as feature selectors using the random forest classifier on wavelet features.**

the data did not follow a normal distribution. Therefore, in the second phase, the authors proceeded to conduct a non-parametric test, namely the Wilcoxon signed-rank test, to determine if the differences between the employed algorithms were statistically significant. The results of the Wilcoxon test are presented in Tables 9–14, and a level of confidence of 95% was used for the statistical analysis. Note that the significance analysis was not done for the accuracy of the algorithms but also for all other performance metrics (precision, recall, F-Score, and MCC and ROC). This would further increase the confidence of selecting an algorithm of the other.

The statistical analysis was done for Scenarios 1–3 (presented above). For these scenarios, both wavelet and curvelet features were employed to study their classification performance. The first scenario is aiming to compare the three feature selection methods (i.e., GA, GWO and PSO-based methods). A comparison of the significance of the three methods was conducted in terms of accuracy, time consumed to determine the set of features and the number of selected features. The aim of this step was to study whether the average performance of the three feature selection methods is equal or not using Wilcoxon test. Table 9 shows that there is no statistically significant difference in accuracy between GAO, GWO, and PSO. Howver, in terms of time consumed, the results showed that there is statistical significance between GWO versus PSO or GAO. Hence, it generally seems that GWO was able to outperform both PSO and GAO when employing wavelet features. Moreover, in the case of the curvelet feature, GWO was better than PSO and GAO in terms of time consumed and the number of features as shown in Table 10.

Scenario 2 aims to study the performance of three different learning strategies (see above). An statistical analysis of their significance, as given in Table 11, shows that there is a statistical

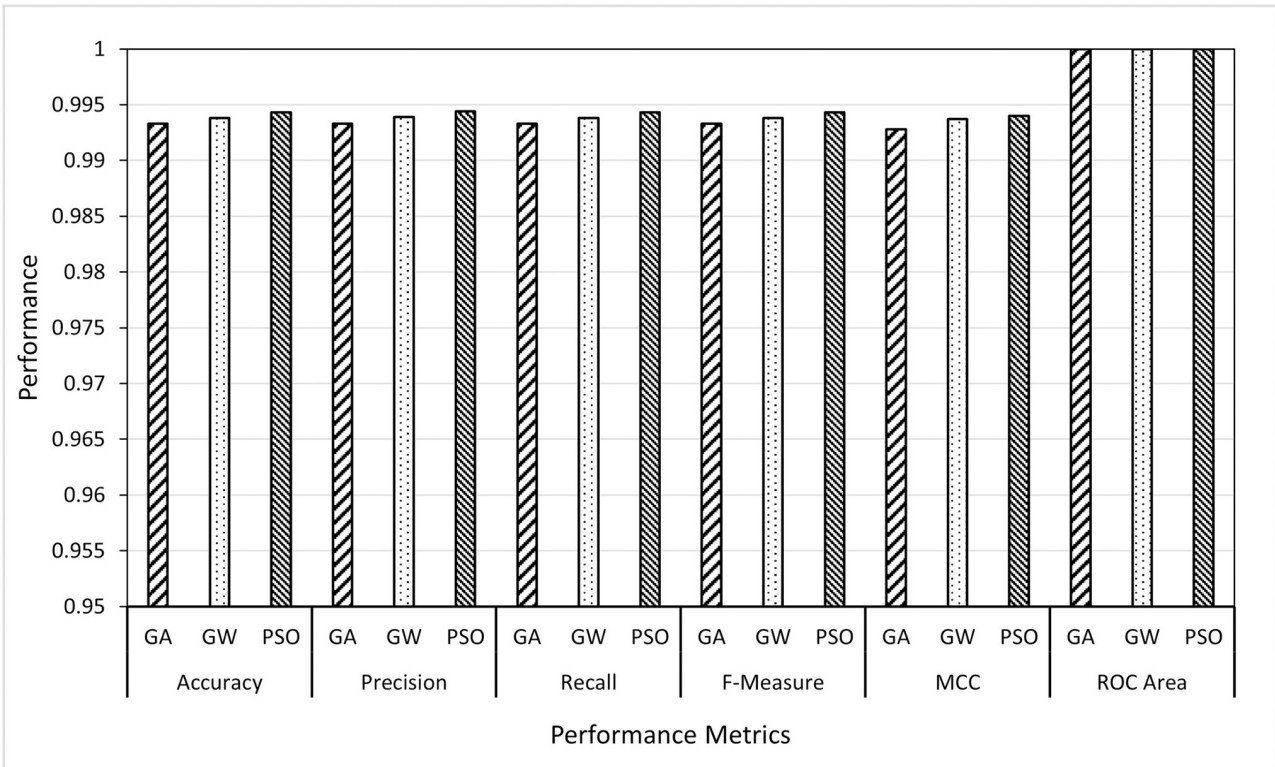

**Fig 8. Comparing GA, GWO and PSO as feature selectors using the random forest classifier on curvelet features.**

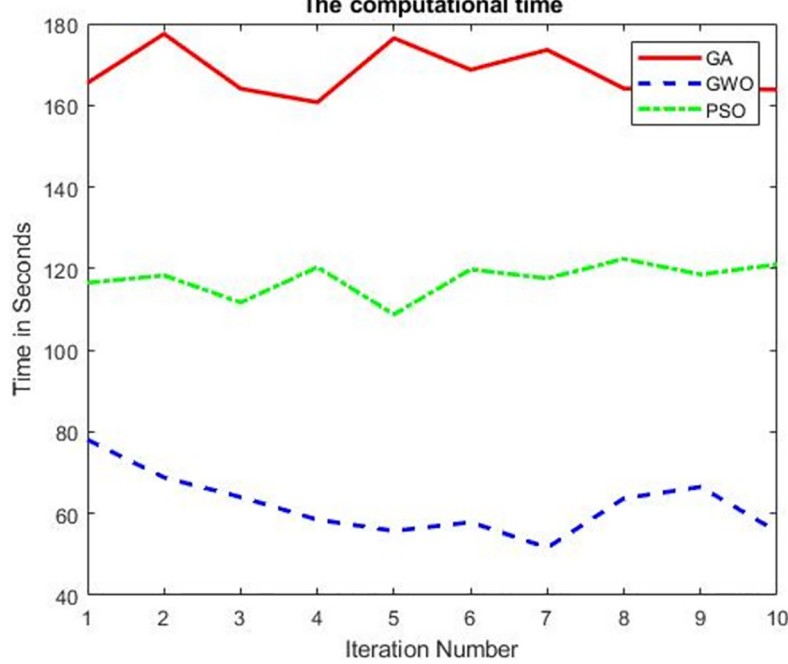

**Fig 9. The computational time between three different feature selection methods GA, GWO and PSO.**

**Table 9. Statistical analysis Scenario 1 using wavelet features.**

| Feature Selection Optimizer | The difference is statistically significant | | | | | |
|---|---|---|---|---|---|---|
| | P-Value | Accuracy | P-Value | Time | P-Value | Number of features |
| GWO Vs. GA | 0.38396 | No | 0.00018267 | Yes | 0.033687 | Yes |
| GAO Vs. PSO | 0.32484 | No | 0.0036105 | Yes | 0.00017761 | Yes |
| GWO Vs. PSO | 0.57061 | No | 0.00018267 | Yes | 0.00017861 | Yes |

**Table 10. Statistical analysis Scenario 1 using curvelet features.**

| Feature Selection Optimizer | The difference is statistically significant | | | | | |
|---|---|---|---|---|---|---|
| | P-Value | Accuracy | P-Value | Time | P-Value | Number of features |
| GWO Vs. GA | 0.25612 | No | 0.00018267 | Yes | 0.00018165 | Yes |
| GAO Vs. PSO | 0.96927 | No | 0.00018267 | Yes | 0.00018063 | Yes |
| GWO Vs. PSO | 0.32282 | No | 0.00018267 | Yes | 0.00018165 | Yes |

**Table 11. Statistical analysis of Scenario 2 using wavelet features.**

| Calssifer | The difference is statistically significant | | | | | | | | | | | |
|---|---|---|---|---|---|---|---|---|---|---|---|---|
| | P-Value | Accuracy | P-Value | Precision | P-Value | Recall | P-Value | F-Measure | P-Value | MCC | P-Value | ROC |
| RF Vs. KNN | 0.013732 | Yes | 0.012346 | Yes | 0.013732 | Yes | 0.01537 | Yes | 0.012379 | Yes | 0.00012698 | Yes |
| KNN Vs. NB | 0.93963 | No | 0.67713 | No | 0.93963 | No | 0.44867 | No | 0.67678 | No | 0.00012934 | Yes |
| RF Vs. NB | 0.0035102 | Yes | 0.0035602 | Yes | 0.0035102 | Yes | 0.0012766 | Yes | 0.0016654 | Yes | 0.81462 | No |

**Table 12. Statistical analysis of Scenario 2 using curvelet features.**

| Calssifer | The difference is statistically significant | | | | | | | | | | | |
|---|---|---|---|---|---|---|---|---|---|---|---|---|
| | P-Value | Accuracy | P-Value | Precision | P-Value | Recall | P-Value | F-Measure | P-Value | MCC | P-Value | ROC |
| RF Vs. KNN | 0.0016252 | Yes | 0.0014201 | Yes | 0.0016252 | Yes | 0.0016252 | Yes | 0.0010652 | Yes | 6.1133e-05 | Yes |
| KNN Vs. NB | 0.00018063 | Yes | 0.00017962 | Yes | 0.00018063 | Yes | 0.00018063 | Yes | 0.00017962 | Yes | 0.13416 | No |
| RF Vs. NB | 0.00017861 | Yes | 0.00017661 | Yes | 0.00017861 | Yes | 0.00017861 | Yes | 0.00017661 | Yes | 6.1133e-05 | Yes |

**Table 13. Statistical analysis of Scenario 3 using wavelet features.**

| Feature Selection Optimizer | The difference is statistically significant | | | | | | | | | | | |
|---|---|---|---|---|---|---|---|---|---|---|---|---|
| | P-Value | Accuracy | P-Value | Precision | P-Value | Recall | P-Value | F-Measure | P-Value | MCC | P-Value | ROC |
| GWO Vs. GAO | 0.39831 | No | 0.42367 | No | 0.35961 | No | 0.40248 | No | 0.34123 | No | 0.30056 | No |
| GWO Vs. PSO | 0.00094304 | Yes | 0.00093881 | Yes | 0.00094304 | Yes | 0.00093881 | Yes | 0.00062385 | Yes | 0.076716 | No |
| PSO Vs. GAO | 0.00019088 | Yes | 0.00025197 | Yes | 0.00025197 | Yes | 0.00025753 | Yes | 0.00028139 | Yes | 0.36812 | No |

**Table 14. Statistical analysis of Scenario 3 using Curvelet features.**

| Feature Selection Optimizer | The difference is statistically significant | | | | | | | | | | | |
|---|---|---|---|---|---|---|---|---|---|---|---|---|
| | P-Value | Accuracy | P-Value | Precision | P-Value | Recall | P-Value | F-Measure | P-Value | MCC | P-Value | ROC |
| GWO Vs. GAO | 0.46009 | No | 0.43682 | No | 0.46009 | No | 0.46009 | No | 0.25973 | No | NaN | N/A |
| GWO Vs. PSO | 0.6091 | No | 0.57858 | No | 0.6091 | No | 0.6091 | No | 0.75699 | No | NaN | N/A |
| PSO Vs. GAO | 0.097034 | No | 0.088674 | No | 0.097034 | No | 0.097034 | No | 0.1264 | No | NaN | N/A |

difference between RF and KNN or NB in the case of using wavelet features. While in the case of using curvelet features, as illustrated in Table 12, the difference is significant between RF against both KNN and NB.

In Scenario 3, the objective was to check the performance of using RF classifier with the different feature selection methods (i.e., GAO, PSO, and GWO-based methods), see above. The statistical analysis of the these results are summarized in Tables 13 and 14. The obtained results show that in the case of using wavelet features, the difference between GWO and PSO is statistically significant and also between the PSO and GAO methods. While in the case of using curvelet features, there is no statistical significance between the three methods since all P-values are greater than 0.05.

## 7 Discussion

In this section, the results reported above will be discussed in terms of the research questions, possible applications, and comparison with most related work.

### 7.1 Research questions discussion

In the introduction, we identified two research questions (RQ1- which is better (wavelet or curvelet transform) in extracting thermal face features to accurately and accurately recognise a user? and RQ2- which metaheuristic algorithms, GWO, GA and PSO, can select the minimum set of features that efficiently and accurately recognise users from their thermal face images?).

For RQ1, the results, reported above section, showed all curvelet-based experiments were better than the results of wavelet-based experiments. In other words, the curvelet transformation was found more effective at producing discriminative features that can accurately describe the curves in thermal face images, which in turn led to higher classification performance in all evaluation metrics. One reason for this improved performance is that curvelet analyzes images from more angles than wavelet, which only analyze images in three directions (horizontal, vertical, and diagonal). Curvelets also have scale, location, and orientation parameters that allow them to tune to various orientations and scales, while wavelets only have scale and location parameters. Additionally, the curvelet covers the entire spectrum in the frequency domain. This means that there is no loss of information when capturing frequency information from the face images. Furthermore, the curvelet is particularly useful for representing singularities over geometric structures in images, whereas wavelet is better suited for point singularities [37]. Overall, the results of the scenarios above suggest that curvelet transforms offer advantages over wavelet transforms when it comes to accurately describing curves in thermal face images and extracting discriminative features for the classification of thermal face images (i.e., authenticating users through their thermal face).

For RQ2, the results obtained from Scenario 3 show two remarks. First, PSO could select a set of features which can achieve slightly higher classification performance than the cases of using GWO and GA. However, this higher performance required more number of features than the case of GWO and GA. Thus, the model with PSO-based features could be adopted by applications where high computational cost is not an issue (e.g., cloud-based face recognition applications). Second, with the minimum set of features, GWO could achieve the second-best classification performance, 99.5% accuracy. This accuracy is less than the PSO result by only 0.02% but GWO results were achieved by features less PSO results with 30%, see Table 8. This means that the model with GWO-based features could be adopted by limited resources applications such as IoT applications.

## 7.2 Possible applications

Thermal face recognition has a wide range of applications in different industries, particularly in situations where traditional face recognition technology may not be effective, such as in low-light or high-contrast environments. In security applications, thermal face recognition with high performance, like the proposed models, can detect and identify individuals wearing masks or other face coverings, which can be difficult for traditional visible light face recognition systems. This makes thermal face recognition an important tool for law enforcement. The wavelet-based model (see Table 8) could be a good candidate for this application as it is lightweight enough to be installed on a portable camera. In addition, thermal face recognition can be used to identify individuals at a distance, which can be useful for security applications in large areas such as airports, train stations, or sports stadiums. The proposed curvelet-based model (see Table 8) would be an ideal solution in such applications where it can be deployed on a cloud or fog computing model.

However, there are some limitations to thermal face recognition technology. For example, it may not be as effective in identifying individuals with similar thermal signatures, such as twins, and it may not be able to accurately identify individuals who have undergone plastic surgery or other facial alterations.

## 7.3 Comparison with related work

We compared the results of the proposed approach with the results of the related works that used the same public dataset (Terravic Facial I) and a summary of this comparison is given in Table 15.

From the table below, two main remarks can be noticed. Firstly, the impact of the proposed GWO-thermal-face-feature selection (GWO-TFFS) on the computation time and the efficiency of the thermal face-based authentication method. Our proposed model is the only model which gave above 99.5% for all evaluation metrics with only 16% of the features space. This would lead to less computational cost which makes our proposed authentication method energy efficient and suitable for IoT applications. Secondly, the suggested model has been thoroughly evaluated using the benchmark evaluation metrics (accuracy, precision, recall, F1-Score, MCC, and ROC curve) but all other related work has only been evaluated using accuracy which is not enough to assess the rigour of the face-based biometric authentication.

A recent work [17] has reported very good results (100% accuracy) but the proposed method not clearly described. It was not clear how the ROI (Face area) was extracted, the purpose of wavelet was not clear, the parameter of SVM and ANN were not specified, finally

**Table 15. Comparison with related work used Terravic Facial IR dataset.**

| Year | Feature Extracted | Feature Selection | Classifiers | Performance |
|---|---|---|---|---|
| [21] 2012 | Haar wavelet,LBP | N/A | ANN Minimum distance | Accuracy -95.09% |
| [14] 2018 | N/A | Rough Set-based | Adaboost | Accuracy-99% |
| [26] 2018 | Zernike moments | N/A | MLPNN | Accuracy 89.5% |
| [18] 2019 | N/A | N/A | HOG-SVM and Back-propagation | HOG-SVM 98.43%, Back-propagation 100% |
| [17] 2022 | Statistical Features | PCA | SVM, ANN Classifier | SVM 99.87% ANN 100% |
| **Proposed system** | Curvelet Features | GWO | RF Classifier | Acc: 99.7% Precision = 99.44% Recall = 99.43% F-Score = 99.43% |

the reduction percentage of feature selection was not reported. Thus, the results seem not reliable.

## 8 Conclusion and future work

This paper proposed a biometric authentication system using thermal images. The system is divided into five phases: (1) capturing a user's face with a thermal camera; (2) segmenting the face region with an optimized superpixel-based segmentation technique; (3) extracting face features with wavelet and curvelet; (4) selecting best features with GWO, PSO, and GA; and (5) classifying or identifying the user with classifiers such as Random Forest, KNN, and Naive Base. The public dataset, Terravic Facial IR, was in the evaluation of the proposed. The results showed that curvelet-based features gave better results than that of wavelet-based features in terms of: accuracy, precision, recall, F-measure, and ROC area. Also, the GWO showed to the best feature selectors among PSO and GA. The GWO was able to select a set of feature less than 20% of the total curevlet features and still gave 99.5% accuracy and other above metrics. Also, it was proved that the ensemble-based learning strategy (i.e., RF) is better than the instance-based learning (i.e., KNN), and and probabilistic-based learning (i.e., NB) in the classification results. In short, it was shown that the GWO-optimized curvelet features of thermal face images can accurately be identified and authenticate users using RF with accuracy, precision, recall, F-measure, and ROC area over 99.5%.

In the future, it is worth to investigate more bio-inspired algorithms from the evolutionary and the swarm classes to build a strong conclusion of which class is better in selecting the most discriminative features for thermal face recognition.

additionally, the proposed models could be investigated under real time conditions. Face recognition in real-time has various applications across a range of industries, including security, retail, and entertainment. It is worthwhile to investigate how thermal face recognition would address the real time visible face recognition challenges including low-memory portable devices, such as microcontrollers which require careful consideration of memory utilisation and the prioritisation of tasks. Also, using different public datasets collected under different conditions would be interesting to see how the proposed models would work and how they can be modified to achieve better results.

The limitations of the study are as follows: (1) the proposed models can take a longer time when tested on big data with millions of instances, and (2) if the problem size is too large, it might not be possible to store the processing data in the memory of the computer running these models.

## Supporting information

**S1 File.**
(TXT)

## Author Contributions

**Conceptualization:** Mohamed Meselhy Eltoukhy, Tarek Gaber.

**Data curation:** Mohamed Meselhy Eltoukhy, Tarek Gaber.

**Formal analysis:** Mohamed Meselhy Eltoukhy, Tarek Gaber.

**Funding acquisition:** Mona A. S. Ali.

**Investigation:** Mona A. S. Ali, Fathimathul Rajeena P. P., Tarek Gaber.

**Methodology:** Mona A. S. Ali, Mohamed Meselhy Eltoukhy, Tarek Gaber.

**Resources:** Mona A. S. Ali, Fathimathul Rajeena P. P.

**Software:** Mohamed Meselhy Eltoukhy, Tarek Gaber.

**Supervision:** Mona A. S. Ali, Tarek Gaber.

**Validation:** Mohamed Meselhy Eltoukhy, Fathimathul Rajeena P. P.

**Writing – original draft:** Mona A. S. Ali, Mohamed Meselhy Eltoukhy, Fathimathul Rajeena P. P., Tarek Gaber.

**Writing – review & editing:** Mona A. S. Ali, Mohamed Meselhy Eltoukhy, Fathimathul Rajeena P. P., Tarek Gaber.

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
