## [Decision Letter · Decision Letter 0]

17 Apr 2023

PONE-D-23-08936Efficient thermal face recognition method using optimized curvelet features

for biometric authenticationPLOS ONE

Dear Dr. Ali,

Thank you for submitting your manuscript to PLOS ONE. After careful consideration, we feel that it has merit but does not fully meet PLOS ONE’s publication criteria as it currently stands. Therefore, we invite you to submit a revised version of the manuscript that addresses the points raised during the review process.

We look forward to receiving your revised manuscript.

Kind regards,

Aytaç Altan, Ph.D.

Academic Editor

PLOS ONE

Journal Requirements:

   "The authors extend their appreciation to the Deputyship for Research and Innovation, the Ministry of Education in Saudi Arabia, for funding this research work (project number INST207"

   "The authors extend their appreciation to the Deputyship for Research and Innovation,the Ministry of Education in Saudi Arabia, for funding this research work (projectnumber INST207)"

  "The authors extend their appreciation to the Deputyship for Research and Innovation, the Ministry of Education in Saudi Arabia, for funding this research work (project number INST207"

Additional Editor Comments:

The reviewers reviewed your manuscript and expressed the opinion that some points should be handled carefully. You should revise your manuscript according to the opinions of the reviewers. In addition to the comments of the reviewers, it is useful to address the following points meticulously:

1) The readability and presentation of the study should be further improved. The paper suffers from language problems.

2) “Discussion” section should be edited in a more highlighting, argumentative way. The author should analysis the reason why the tested results is achieved. It will be helpful to the readers if some discussions about insight of the main results are added as Remarks.

3) How to set the parameters of proposed method for better performance?

4) The significance of the design carried out in this paper is not well explained relative to other important works published in this field. The authors should review, comment, and compare more works that are developed recently. The authors should clearly emphasize the contribution of the study. Please note that the up-to-date of references will contribute to the up-to-date of your manuscript. The study named " Recognition of COVID-19 disease from X-ray images by hybrid model consisting of 2D curvelet transform, chaotic salp swarm algorithm and deep learning technique; Artificial intelligence-based robust hybrid algorithm design and implementation for real-time detection of plant diseases in agricultural environments"- can be used to explain the object detection process in the study.

5) The main contributions of the study can be given in the last paragraph of the Introduction section.

Reviewers' comments:

Reviewer's Responses to Questions

**Comments to the Author**

1. Is the manuscript technically sound, and do the data support the conclusions?

Reviewer #1: Yes

Reviewer #2: Yes

Reviewer #3: Yes

2. Has the statistical analysis been performed appropriately and rigorously? 

Reviewer #1: Yes

Reviewer #2: No

Reviewer #3: Yes

3. Have the authors made all data underlying the findings in their manuscript fully available?

Reviewer #1: Yes

Reviewer #2: No

Reviewer #3: Yes

4. Is the manuscript presented in an intelligible fashion and written in standard English?

Reviewer #1: Yes

Reviewer #2: Yes

Reviewer #3: Yes

5. Review Comments to the Author

Reviewer #1: Dear Authors,

although proposed manuscript has merits, there are some issues that need to be addressed.

My observations are below.

- Method names should not be capitalized. Moreover, it is not the best practice to employ abbreviations in the abstract, they should be used when the term is introduced for the first time.

- Introduction should be clearly presented to highlight main ideas and motivation behind the proposed research.

Please include and clearly state research question and contributions of proposed study in Introduction. Also, please clearly explain what is "beyond state-of-the-art" in the proposed study.

- Literature review should be enhanced to include more metaheurisitcs approaches for computer vision, here are some suggested sources:

https://www.sciencedirect.com/science/article/pii/S0045790622003172

https://www.nature.com/articles/s41598-022-09744-2

- Visualization of results should be improved - consider using box and whiskers diagrams, swarm plots, etc. Also, please include convergence speed graphs, since the convergence is very important indicator of metaheuristics performance.

It would be interesting how population diversity of proposed method changes during the course of iterations of the best run, this can be easily done by using e.g. swarm plot diagrams.

- Although statistical tests are performed well, you did not justify we you have employed ANOVA. First you should conduct, for example Shapiro Wilk test to see whether or not results from different runs come from the normal distribution and to check if all three conditions for safe use of parametric tests are satisfied. After that you should proceed with parametric or non-parametric tests.

a. conduct a normality test, then

b. if the variable follows a normal distribution, the average and standard deviation should be complemented with a parametric test for significant difference;

c. otherwise, you should conduct a non-parametric test and a post hoc analysis.

- Conclusion should be extended to include more details regarding the future work and limitations of proposed study.

- Some references are missing parts, such as pp., publisher, year, etc.

- There are some English language and technical errors, please revise, e.g. tables must be aligned on the center of the page, all symbols in equations must be defined, etc.

Reviewer #2: Reviewer comments:

The paper "Efficient thermal face recognition method using optimized curvelet features for biometric authentication" presents a thermal face-based biometric authentication system that uses optimized curvelet features and bio-inspired optimization algorithms. The proposed system is evaluated using several performance metrics and compared with related works. Overall, the paper presents a well-written and organized study with valuable contributions to the field of biometric authentication. However, there are some points that need to be addressed:

The paper lacks a clear problem statement and research gap. The introduction section provides general information about biometric authentication and thermal imaging, but it is not clear what specific problem the proposed system aims to solve or how it contributes to the existing literature. The authors should provide a more focused and concise problem statement and explain how their proposed system fills a research gap.

The paper would benefit from more detailed explanations of the methods used. While the authors provide an overview of the system and its phases, the methods used in each phase are not described in sufficient detail. For example, it is unclear how the bio-inspired optimization algorithms are applied to feature selection or how the classifiers are trained. The authors should provide more detailed explanations of the methods used and the reasoning behind their choices.

The paper could benefit from a more thorough evaluation of the proposed system. While the authors present several performance metrics and compare their results with related works, they do not provide a detailed analysis of the results or discuss the limitations of the proposed system. For example, it is unclear how the system performs on different datasets or in different scenarios. The authors should provide a more thorough evaluation of the proposed system and discuss its limitations.

The paper could benefit from more discussion of the implications and applications of the proposed system. While the authors briefly mention some potential applications of thermal face-based biometric authentication, they do not provide a detailed discussion of the implications or limitations of the proposed system. The authors should provide more detailed discussion of the implications and applications of their proposed system.

Feedback to improve:

Provide a more focused and concise problem statement that clearly states the research gap and the specific problem the proposed system aims to solve.

Provide more detailed explanations of the methods used, including how the bio-inspired optimization algorithms are applied to feature selection and how the classifiers are trained.

Provide a more thorough evaluation of the proposed system, including a detailed analysis of the results and discussion of the limitations of the proposed system. Also, try to provide some analysis on the complexity of the proposed method in comparison with the others.

Provide more detailed discussion of the implications and applications of the proposed system, including its potential applications and limitations.

Reviewer #3: The article titled “Efficient thermal face recognition method using optimized curvelet features for biometric authentication” has been studied in detail. In summary, in the study, feature selection with optimization algorithms, comparison of algorithms, comparison of classifiers and statistical analyzes were made to perform face recognition operations. The topic covered in the article is remarkable and the article contains useful information. However, there are some problems that need to be addressed by the authors:The introduction can be expanded to provide a clearer context for why the future of the sharing economy is be predicted using this and similar techniques:

1) Is “ANOVA” mentioned in the summary an abbreviation? I recommend stating that it refers to statistical analysisAre the datasets used in the study public or private? It must be specified.

2) In the experiment, there is no fitness function, which is important in terms of computational cost in the feature selection process of optimization algorithms. I would recommend specifying which fitness function is used in the article and why it is used.

3) Why is the db4 family preferred only as the wavelet family in the features extracted with wavelet? Why were other families not included in the study?It would be nice if the code of the algorithm could be shared.

4) It has not been clearly stated in the article for how many levels the wavelet transform is performed. I recommend specifying it.

5) How many of the images in the dataset were used for the testing of the model and how much for the training of the model? Please specify.

6) Has the algorithm been tested in real time? If yes, are the computational costs compatible with the simulation study?

6. PLOS authors have the option to publish the peer review history of their article (what does this mean?). If published, this will include your full peer review and any attached files.

Reviewer #1: No

Reviewer #2: No

Reviewer #3: No

---

## [Author Response · Author response to Decision Letter 0]

16 May 2023

Response to Editor

Article ID: PONE-D-23-08936

Original Article Title: Efficient thermal face recognition method using optimized curvelet features for biometric authentication

Re: Response to Editor

Dear Editor-in-chief,

Thank you for allowing a resubmission of our manuscript, with an opportunity to address the reviewers’ comments. We appreciate the Editor and the reviewers for taking the time to review our manuscript carefully and for providing valuable feedback to help us improve the quality of the document.

We have made every effort to eliminate all the indicated major and minor inconsistencies and do hope that as a result, the quality of our revised manuscript has improved further. The updated parts are highlighted in red color.

Point 1: The readability and presentation of the study should be further improved. The paper suffers from language problems 

Response 1: We appreciate this comment. We restructured and improved the flow of the paragraphs of the paper to improve its readability and presentation. We also carefully proofread the entire paper

Point 2: “Discussion” section should be edited in a more highlighting, argumentative way. The author should analysis the reason why the tested results is achieved. It will be helpful to the readers if some discussions about insight of the main results are added as Remarks.

Response 2: Thanks for pointing out this important issue. We added 3 subsections in the discussion. Section, 7.1 Research Questions Discussion which discusses the results in terms of two research questions in an argumentative way and identifies Remarks from this discussion. In Section, 7.2, we discussed the possible applications of the proposed models while in Section 7.3, we compared and discussed our model with the most related work

Point 3: How to set the parameters of proposed method for better performance?

Response 3: The parameter's best values were identified in presented in Tables 2-7.

Point 4: The significance of the design carried out in this paper is not well explained relative to other important works published in this field. The authors should review, comment, and compare more works that are developed recently. The authors should clearly emphasize the contribution of the study. Please note that the up-to-date of references will contribute to the up-to-date of your manuscript. The study named " Recognition of COVID-19 disease from X-ray images by hybrid model consisting of 2D curvelet transform, chaotic salp swarm algorithm and deep learning technique; Artificial intelligence-based robust hybrid algorithm design and implementation for real-time detection of plant diseases in agricultural environments"- can be used to explain the object detection process in the study.

Response 4: Thank you so much for your valuable comment. Responding to this comment, we did the following: 

1- cited the recommended articles, See lines 40-40

 2- added the significance, the challenge and the gap in knowledge of the study in the introduction section, See lines 64-83

3- The contribution of the study has been re-stated and emphasized, See lines 79-118

Point 5: The main contributions of the study can be given in the last paragraph of the Introduction section.

Response 5: Thank you so much for your valuable comment. 

The contribution of the study has been re-stated and emphasized, See lines 79-118 

Response to Reviewer 1

Article ID: PONE-D-23-08936

Original Article Title: Efficient thermal face recognition method using optimized curvelet features for biometric authentication

Re: Response to reviewer 1

Dear Reviewer 1,

We, the authors, would like to express our sincere gratitude for providing an opportunity to revise our manuscript and improve our work. We have worked out the comments to the best of our knowledge. With the constructive comments offered, we are sure that the revised manuscript will be of better quality. The authors have carefully addressed the comments and proofread the entire article for language and grammar correction. The corrections are marked as red in the revised manuscript and described in our response in colored text. Some of the modifications are common. We hope that our modified work will the considered. Once again, thank you for providing an opportunity to improve our work.

Comments and Responses

Comment 1- Method names should not be capitalized. Moreover, it is not the best practice to employ abbreviations in the abstract, they should be used when the term is introduced for the first time.

Response 1: Thank you so much for your comment. All the abbreviations have been introduced before using them and the abstract is updated too. The whole document has been checked and modified. A table of abbreviations is added after the conclusion 

Comment 2- Introduction should be clearly presented to highlight main ideas and motivation behind the proposed research.

 Please include and clearly state research question and contributions of proposed study in Introduction. Also, please clearly explain what is "beyond state-of-the-art" in the proposed study.

Response 2: Thank you so much for your valuable comment. Responding to this comment, we did the following: 

1- added the motivation of the study, See lines 2-22

 2- added the significance, the challenge and the gap in knowledge of the study in the introduction section, See lines 64-78

3- added two research questions, See lines 79-83

4- The contribution of the study has been re-stated and emphasized, See lines 79-118 

Comment 3- Literature review should be enhanced to include more metaheurisitcs approaches for computer vision, here are some suggested sources:

https://www.sciencedirect.com/science/article/pii/S0045790622003172

https://www.nature.com/articles/s41598-022-09744-2

Response 3: These references have been cited as [5] and [7] in the revised version

Comment 4- Visualization of results should be improved - consider using box and whiskers diagrams, swarm plots, etc. Also, please include convergence speed graphs, since the convergence is very important indicator of metaheuristics performance.

It would be interesting how population diversity of proposed method changes during the course of iterations of the best run, this can be easily done by using e.g. swarm plot diagrams.

Response 4: Thank you for the comment. We have added the required part under the analysis and discussion section in the document

The plot of the best fitness value versus the number of iterations is a way to visualise the performance of an optimization algorithm. The x-axis represents the number of iterations, while the y-axis represents the best fitness value achieved by the algorithm at that iteration. The plots are showing a decreasing trend in the fitness value as the number of iterations increases, indicating that the algorithm is making progress towards finding a good solution. In this work, the fitness function is the accuracy of the model on the dataset. The goal is to maximise the accuracy, so the fitness function would return a higher value for better models.

Comment 5- Although statistical tests are performed well, you did not justify we you have employed ANOVA. First you should conduct, for example Shapiro Wilk test to see whether or not results from different runs come from the normal distribution and to check if all three conditions for safe use of parametric tests are satisfied. After that you should proceed with parametric or non-parametric tests.

a. conduct a normality test, then

b. if the variable follows a normal distribution, the average and standard deviation should be complemented with a parametric test for significant difference;

c. otherwise, you should conduct a non-parametric test and a post hoc analysis.

Response 5: 

The section of statistical analysis has been updated: 

The statistical analysis was carried out in two different steps. In the first step, we investigated whether the data followed a normal distribution or not, using the Shapiro-Wilk test. The analysis showed that the data did not follow a normal distribution. Therefore, we conducted a calculation to calculate if the difference is significant using a non-parametric test, namely the Wilcoxon test. The results of the wilcoxon test are presented in Tables 9-14. The statistical analysis was used to evaluate the performance of the proposed methods with a level of confidence of 95%.

Comment 6- Conclusion should be extended to include more details regarding the future work and limitations of proposed study.

Response 6: We appreciate this comment. We added future work and limitations of the study in the conclusion section( lines 718-729) .

In the future, the proposed models could be investigated under real-time conditions. Face recognition in real-time has various applications across a range of industries, including security, retail, and entertainment. It is worthwhile to investigate how thermal face recognition would address the real-time visible face recognition challenges, including low-memory portable devices, such as microcontrollers which require careful consideration of memory utilisation and the prioritisation of tasks. Also, using different public datasets collected under different conditions would be investigated to see how the proposed models would work and how they could be modified to achieve better results. 

The limitations of the study are as follows: (1) the proposed models can take a longer time when tested on big data with millions of instances, and (2) if the problem size is too large, it might not be possible to store the processing data in the memory of the computer running these models.

Comment 7- Some references are missing parts, such as pp., publisher, year, etc.

Response 7: Thank you for pointing out the mistakes in references. All references have been checked and edited accordingly.

Comment 8- There are some English language and technical errors, please revise, e.g. tables must be aligned on the center of the page, all symbols in equations must be defined, etc.

Response 8: The whole document has been proofread for language and errors . 

Response to Reviewer 2

Article ID: PONE-D-23-08936

Original Article Title: Efficient thermal face recognition method using optimized curvelet features for biometric authentication

Re: Response to reviewers

Dear Reviewer 2,

Thank you for allowing a resubmission of our manuscript, with an opportunity to address the reviewers’ comments. We appreciate the Editor and the reviewers for taking the time to review our manuscript carefully and for providing valuable feedback to help us improve the quality of the document.

We have made every effort to eliminate all the indicated major and minor inconsistencies and do hope that as a result, the quality of our revised manuscript has improved further. The updated parts are highlighted in red color.

Point 1: The paper lacks a clear problem statement and research gap. The introduction section provides general information about biometric authentication and thermal imaging, but it is not clear what specific problem the proposed system aims to solve or how it contributes to the existing literature. The authors should provide a more focused and concise problem statement and explain how their proposed system fills a research gap. 

Provide a more focused and concise problem statement that clearly states the research gap and the specific problem the proposed system aims to solve.

Response1: Thank you so much for your valuable comment. Responding to this comment, we did the following: 

1- add the motivation of the study, See lines 2-22

 2- added the significance, the challenge and the gap in knowledge of the study in the introduction section, See lines 64-78

3- added two research questions, See lines 79-83

4- The contribution of the study has been re-stated and emphasized, See lines 79-118

Point 2: The paper would benefit from more detailed explanations of the methods used. While the authors provide an overview of the system and its phases, the methods used in each phase are not described in sufficient detail. For example, it is unclear how the bio-inspired optimization algorithms are applied to feature selection or how the classifiers are trained. The authors should provide more detailed explanations of the methods used and the reasoning behind their choices. 

Provide more detailed explanations of the methods used, including how the bio-inspired optimization algorithms are applied to feature selection and how the classifiers are trained.e proposed study.

Response 2: Thank you so much for your valuable comment. Responding to this comment, we did the following: 

1- Give a full section (4 Proposed Approach) for the proposed system and its phases, supported by Figure 1. 

2- To also show how the bio-inspired optimization algorithms are applied to feature selection as well as how the population diversity of the proposed method changes during the course of iterations to find the best solution, we added the pseudo-code for algorithms (GA, PSO, and GWO) in Section 3.3-3.5. 

3- For the training of the classifiers, we added more explanation and justification (see Section 5.3, Lines 493-511) 

4- We added more discussion (Section 7.1 and 7.2) Page 26&27

Point 3: The paper could benefit from a more thorough evaluation of the proposed system. While the authors present several performance metrics and compare their results with related works, they do not provide a detailed analysis of the results or discuss the limitations of the proposed system. For example, it is unclear how the system performs on different datasets or in different scenarios. The authors should provide a more thorough evaluation of the proposed system and discuss its limitations. 

Provide a more thorough evaluation of the proposed system, including a detailed analysis of the results and discussion of the limitations of the proposed system. Also, try to provide some analysis on the complexity of the proposed method in comparison with the others.

Response 3: Thank you so much for your valuable comment. Responding to this comment, we did the following: 

1- A detailed analysis of the results was given (see pages-18-20)

2- We added more discussion (Sections 7.1 and 7.2), see Pages 26&27

3- The limitations of the proposed models are given at the end of two sections, Section, 7.2 Possible Applications, and the “conclusion and future work” section. 

4- The time complexity of the algorithms was computed. See Section 6.4 and Figure 9.

5- Further analysis of the convergence of the employed bioinspired algorithms for the feature selection problem was in given Figures 2, 3, & 4, pages 20-21..

Point 4: The paper could benefit from more discussion of the implications and applications of the proposed system. While the authors briefly mention some potential applications of thermal face-based biometric authentication, they do not provide a detailed discussion of the implications or limitations of the proposed system. The authors should provide more detailed discussion of the implications and applications of their proposed system. 

Provide more detailed discussion of the implications and applications of the proposed system, including its potential applications and limitations.

Response 4: Thank you so much for your valuable comment. Responding to this comment, we did the following: 

1- A new section (7.2 Possible Applications) was created to extend and discuss the possible applications and implications of the thermal face-based recognition system, See page 27.

2- As explained above, the limitations of the proposed models are given at the end of two sections, Section, 7.2 Possible Applications, and the “conclusion and future work” section. Also, a detailed analysis of the results was given (see pages-18-20) and in Sections 7.1, a new section.

3- The limitations of the proposed models are given at the end of two sections, Section, 7.2 Possible Applications, and the “conclusion and future work” section. 

 4- A comparison with Related Work was given in Section 7.3 

Response to Reviewer 3

Article ID: PONE-D-23-08936

Original Article Title: Efficient thermal face recognition method using optimized curvelet features for biometric authentication

Re: Response to reviewers

Dear Reviewer 3,

Thank you for allowing a resubmission of our manuscript, with an opportunity to address the reviewers’ comments. We appreciate the Editor and the reviewers for taking the time to review our manuscript carefully and for providing valuable feedback to help us improve the quality of the document.

We have made every effort to eliminate all the indicated major and minor inconsistencies and do hope that as a result, the quality of our revised manuscript has improved further. The updated parts are highlighted in red color.

Point 1: The introduction can be expanded to provide a clearer context for why the future of the sharing economy is be predicted using this and similar techniques:

Response 1: Thank you so much for your comment. A paragraph is added at the beginning of the Introduction section Lines 2 -12

Facial recognition technology has become a topic of significant interest in recent years, with wide-ranging applications in fields such as security, surveillance, access control, and human-computer interaction. The rapid advancements in computer vision, machine learning, and deep learning have led to significant improvements in the accuracy and reliability of facial recognition systems, making them a promising solution for real-world scenarios.

According to the International Biometrics + Identity Association (IBIA), facial recognition is one of the most widely used biometric modalities, with an estimated market size of 7.76 billion US dollars by 2025, with a compound annual growth rate (CAGR) of 15.3\\% from 2020 to 2025 

The increased adoption of facial recognition systems can be attributed to their non-intrusive nature, ability to operate in real-time, and potential for high accuracy.

Point 2: Is “ANOVA” mentioned in the summary an abbreviation? I recommend stating that it refers to statistical analysis Are the datasets used in the study public or private? It must be specified. 

Provide more detailed explanations of the methods used, including how the bio-inspired optimization algorithms are applied to feature selection and how the classifiers are trained.e proposed study.

Response 2: Appreciated your comment. 

The ANOVA (ANalysis Of VAriance) is a name of a statistical analysis tool. In the revised version, we changed this tool to another tool called Wilcoxon, as required by another reviewer. 

The dataset used in the study is a public dataset called, Terravic Facial IR. This has been mentioned in the abstract and in the data description section. Section (5.1 Dataset)

Point 3: In the experiment, there is no fitness function, which is important in terms of computational cost in the feature selection process of optimization algorithms. I would recommend specifying which fitness function is used in the article and why it is used.

Response 3: Thank you so much for your valuable comment. Responding to this comment, we did the following:

More descriptions of the feature selection method have been presented in Sections 3.3, 3.4 and 3.5. We have presented the method description and Pseudocode illustrating how we employed each feature selection algorithm for the feature selection method. Moreover, presenting the fitness function (the accuracy) and why it was used in Algorithms 1,2 and 3 on pages 10, 11, 12 respectively.

Point 4: Why is the db4 family preferred only as the wavelet family in the features extracted with wavelet? Why were other families not included in the study? It would be nice if the code of the algorithm could be shared.

Response 4: Thank you so much for your valuable comment. Responding to this comment, we did the following:

The efficiency of the Daubechies 4 (DB4) wavelet has been established according to the findings in referenced [38, 39]. As a result, we utilized this wavelet in our proposed technique to decompose the ROI image into four distinct levels. See line 414 Page 14

The code of the Proposed method is ready to share with the journal editor once it is needed.

Point 5: It has not been clearly stated in the article for how many levels the wavelet transform is performed. I recommend specifying it.

Response 5: Thank you so much for your valuable comment. 

The proposed approach was used to decompose the ROI image into four distinct levels, as presented earlier in References [38] and [39]. we have specified the details in the paper line 417 Page 14.

Point 6: Conclusion should be extended to include more details regarding the future work and limitations of proposed study.

Response 6: Thank you so much for your valuable comment. Responding to this comment, we did the following:

 For the training of the classifiers, we added more explanation and justification (see Section 5.3, Lines 493-511) 

A good training approach for a model with a dataset must be found. The model should have enough instances to train on without over-fitting it, but if there aren’t enough, the model won’t be properly trained and will perform poorly when tested [43] and [44]. Using the k-Fold Cross-Validation, with a large value of k (i.e., k approaching the number of instances in the dataset) or leave-one-out cross-validation can help ensure that the model is evaluated on as much of the data as possible. This can provide a more accurate estimate of the model’s performance than a simple train-test split, as each data point is used for both training and validation [ 43 ]. It was justified in [ 43 ] that the k-fold cross-validation is better to be used over hold-out validation. Therefore, in our case, k-fold cross-validation has been used to evaluate the performance of the proposed models. 

Unlike, splitting the dataset into 70% for training and 30% for testing a model. In k-fold cross-validation splitting the dataset into k subsets, or folds, of approximately equal size. The model is then trained on k-1 folds, and the remaining fold is used for validation/testing. This process is repeated k times, with each fold used as the validation set exactly once. The results of the k-folds are then averaged to obtain an overall estimate of the model’s performance. This approach is useful because it allows for a more accurate estimation of the model’s performance, as each data point is used both for training and validation at some point in the process [43] and [44].

[43]. S. Yadav and S. Shukla, "Analysis of k-Fold Cross-Validation over Hold-Out Validation on Colossal Datasets for Quality Classification," 2016 IEEE 6th International Conference on Advanced Computing (IACC), Bhimavaram, India, 2016, pp. 78-83, doi: 10.1109/IACC.2016.25.

[44] Kohavi, Ron. A study of cross-validation and bootstrap for accuracy estimation and model selection. Proceedings of the 14th international joint conference on Artificial intelligence—Volume 2. 1995; p. 1137–1143. “5.3 Experiments Environment Setting”, lines 493-511

Point 7: Has the algorithm been tested in real time? If yes, are the computational costs compatible with the simulation study?

Response 7: Thank you so much for your valuable comment. This is a very interesting point, which we included in the conclusion and Future Work section. This would need a different methodology for using hardware and software to be set in a real environment. Conclusion section

---

## [Decision Letter · Decision Letter 1]

5 Jun 2023

Efficient thermal face recognition method using optimized curvelet features

for biometric authentication

PONE-D-23-08936R1

Dear Dr. Ali,

We’re pleased to inform you that your manuscript has been judged scientifically suitable for publication and will be formally accepted for publication once it meets all outstanding technical requirements.

Kind regards,

Aytaç Altan, Ph.D.

Academic Editor

PLOS ONE

Additional Editor Comments (optional):

Reviewers' comments:

Reviewer's Responses to Questions

**Comments to the Author**

1. If the authors have adequately addressed your comments raised in a previous round of review and you feel that this manuscript is now acceptable for publication, you may indicate that here to bypass the “Comments to the Author” section, enter your conflict of interest statement in the “Confidential to Editor” section, and submit your "Accept" recommendation.

Reviewer #1: All comments have been addressed

Reviewer #3: All comments have been addressed

2. Is the manuscript technically sound, and do the data support the conclusions?

Reviewer #1: Yes

Reviewer #3: Yes

3. Has the statistical analysis been performed appropriately and rigorously? 

Reviewer #1: Yes

Reviewer #3: Yes

4. Have the authors made all data underlying the findings in their manuscript fully available?

Reviewer #1: Yes

Reviewer #3: Yes

5. Is the manuscript presented in an intelligible fashion and written in standard English?

Reviewer #1: Yes

Reviewer #3: Yes

6. Review Comments to the Author

Reviewer #1: Dear Authors,

thank you for addressing all my comments.

According to my opinion, you have carefully revised your manuscript.

There are still minor English/spelling errors, however they will be hopefully corrected during the proofreading.

Warmest

Reviewer #3: The mentioned issues have been discussed in detail by the authors. The paper may be accepted for publication. Also, the code of the model can be shared.

7. PLOS authors have the option to publish the peer review history of their article (what does this mean?). If published, this will include your full peer review and any attached files.

Reviewer #1: No

Reviewer #3: No

---

## [Editor Report · Acceptance letter]

15 Jun 2023

PONE-D-23-08936R1 

Efficient thermal face recognition method using optimized curvelet features
for biometric authentication 

Dear Dr. Ali:

I'm pleased to inform you that your manuscript has been deemed suitable for publication in PLOS ONE. Congratulations! Your manuscript is now with our production department. 

Kind regards, 

on behalf of

Dr. Aytaç Altan 

Academic Editor

PLOS ONE